# Synthesis of lipid-linked precursors of the bacterial cell wall is governed by a feedback control mechanism in *Pseudomonas aeruginosa*

Lindsey S. Marmont[1,2], Anna K. Orta [3], Becca W. A. Baileeves [4,5], David Sychantha [2], Ana Fernández-Galliano [2], Yancheng E. Li [3], Neil G. Greene[1,8], Robin A. Corey[6], Phillip J. Stansfeld [4], William M. Clemons Jr[3] & Thomas G. Bernhardt [1,7] ✉

Many bacterial surface glycans such as the peptidoglycan (PG) cell wall are built from monomeric units linked to a polyprenyl lipid carrier. How this limiting carrier is distributed among competing pathways has remained unclear. Here we describe the isolation of hyperactive variants of *Pseudomonas aeruginosa* MraY, the enzyme that forms the first lipid-linked PG precursor. These variants result in the elevated production of the final PG precursor lipid II in cells and are hyperactive in vitro. The activated MraY variants have substitutions that map to a cavity on the extracellular side of the dimer interface, far from the active site. Our structural and molecular dynamics results suggest that this cavity is a binding site for externalized lipid II. Overall, our results support a model in which excess externalized lipid II allosterically inhibits MraY, providing a feedback mechanism that prevents the sequestration of lipid carrier in the PG biogenesis pathway.

Bacterial cells surround themselves with a complex envelope essential for their integrity and shape. The envelopes of gram-negative (diderm) bacteria also serve as a formidable barrier against the entry of drug molecules, providing organisms such as *Escherichia coli* and *Pseudomonas aeruginosa* with a relatively high intrinsic resistance to antibiotics[1,2]. Understanding how these bacteria construct their envelope and regulate the assembly process therefore promises to aid in the identification of new vulnerabilities in surface biogenesis to target for antibiotic development.

The diderm envelope consists of two membranes: a cytoplasmic (inner) membrane and an asymmetric outer membrane (OM) with an inner leaflet of phospholipids and an outer leaflet composed of lipopolysaccharide (LPS)[2]. The LPS molecule consists of a lipid A moiety, a core oligosaccharide and a long polysaccharide chain called the O-antigen (O-Ag) that varies in composition between different strains and species[3]. Between the inner and outer membranes is the periplasmic space where the peptidoglycan (PG) cell wall is assembled. The PG layer is constructed from glycan strands with repeating

[1]Department of Microbiology, Blavatnik Institute, Harvard Medical School, Boston, MA, USA. [2]Michael DeGroote Institute for Infectious Disease Research, David Braley Centre for Antibiotic Discovery, Department of Biochemistry and Biomedical Sciences, McMaster University, Hamilton, Ontario, Canada. [3]Division of Chemistry and Chemical Engineering, California Institute of Technology, Pasadena, CA, USA. [4]School of Life Sciences and Department of Chemistry, University of Warwick, Coventry, UK. [5]Warwick Medical School, University of Warwick, Coventry, UK. [6]School of Physiology, Pharmacology, and Neuroscience, University of Bristol, Bristol, UK. [7]Howard Hughes Medical Institute, Chevy Chase, MD, USA. [8]Present address: Department of Cell and Molecular Biology, The University of Rhode Island, Kingston, RI, USA. ✉e-mail: thomas_bernhardt@hms.harvard.edu

**Fig. 1 | MraY(T23P) restores growth to strains defective in PG biosynthesis.**
**a,c**, Schematic representation of the aPBPs and their outer membrane lipoprotein activators in *Pseudomonas aeruginosa* (**a**) and *Escherichia coli* (**c**). **b,d**, Tenfold serial dilutions of cells of the indicated *P. aeruginosa* (**b**) or *E. coli* (**d**) strains harbouring expression plasmids producing the indicated MraY variant. Dilutions were plated on the indicated medium with or without IPTG to induce the production of MraY variants as indicated. Asterisks indicate the activated form of *E. coli* PBP1b. Dashed outlines in **a** and **c** represent proteins that are absent in the specified strain. IM, inner membrane; GT, glycosyltransferase; TP, transpeptidase.

N-acetylglucosamine (GlcNAc) and N-acetylmuramic acid (MurNAc) sugars that are crosslinked by peptide stems attached to MurNAc forming the interconnected meshwork that encases the inner membrane[4].

Surface glycans such as PG and O-Ag are polymerized from monomeric building blocks attached to polyprenyl lipids via a pyrophosphate linkage. The lipid carrier is regenerated during polymerization, making it available for the continued production of monomer units to support synthesis of growing polymers[5,6]. This synthetic strategy is conserved throughout biology, with uses ranging from surface glycan biogenesis in microbes to the production of N-linked glycans in eukaryotic cells[7–10]. In any given organism, a common polyprenyl lipid carrier is used to build multiple different glycans[7]. The concentration of these carriers is limiting[8], suggesting that their utilization to produce monomer units for different pathways must be coordinated with the corresponding glycan polymerization process. Otherwise, excess accumulation of monomer units for one polymer will sequester the limiting carrier, indirectly inhibiting the production of other glycans that require the carrier for their biogenesis. Such precursor sequestration can have considerable detrimental consequences for the cell envelope[8–11]. Despite the importance of efficient carrier utilization for the balanced synthesis of different surface glycans, the underlying mechanism has remained elusive.

The membrane-anchored precursor for PG biosynthesis is lipid II. Its synthesis begins in the cytoplasm, where multiple enzymes assemble uridine diphosphate-MurNAc-pentapeptide (simplified as UM5)[4]. The phospho-MurNAc-pentapeptide moiety from this intermediate is then transferred to the lipid carrier undecaprenyl phosphate (C55P) at the inner face of the cytoplasmic membrane by the essential integral membrane enzyme MraY, generating the penultimate PG precursor, lipid I. The peripheral membrane enzyme MurG then transfers GlcNAc from UDP-GlcNAc to lipid I, forming lipid II, which contains the basic monomeric unit of PG. Following its synthesis, lipid II is transported across the cytoplasmic membrane by MurJ where it can then be polymerized and crosslinked by PG synthases to form the cell wall matrix[12].

This investigation started with the study of *P. aeruginosa* mutants with a conditionally lethal defect in the activity of cell wall synthases called class A penicillin-binding proteins (aPBPs)[13]. We isolated suppressors encoding an altered MraY enzyme with a T23P substitution [MraY(T23P)] that restored the growth of these cells in the non-permissive condition. Our characterization of this and other related MraY variants supports a model in which MraY is feedback inhibited by the accumulation of flipped lipid II, limiting the synthesis of PG precursors when their supply exceeds the synthetic capacity of PG synthases.

## Results

### An MraY variant rescues a lethal aPBP synthase defect

*P. aeruginosa* produces two aPBPs, *Pa*PBP1a and *Pa*PBP1b, encoded by the *ponA* and *ponB* genes, respectively. These PG synthases require cognate OM lipoprotein activators to function properly[13,14]. PBP1a is activated by *Pa*LpoA and *Pa*PBP1b is activated by *Pa*LpoP[13] (Fig. 1a). A ΔponB ΔlpoA mutant relies on an unactivated PBP1a enzyme for growth (Fig. 1a). We therefore refer to the strain as a PBP1a-only mutant for simplicity. Such mutants are viable on rich medium (lysogeny broth, LB) with some lysing cells observed[13] (Extended Data Fig. 1), but have severe growth defects on Vogel-Bonner minimal medium (VBMM)[13]. Spontaneous suppressors supporting growth of the PBP1a-only mutant on VBMM were isolated to uncover new insights into PG synthesis regulation. Several of these mutants encoded variants of *Pa*PBP1a, and we previously reported that they bypass the *Pa*LpoA requirement for *Pa*PBP1a function by activating the PG synthase[15]. Thus, the growth defect of the PBP1a-only strain on VBMM is caused by a deficit of aPBP activity. Here we report the identification of another class of suppressor with a mutation in *Pa*mraY encoding an enzyme variant with a T23P substitution (*Pa*MraY(T23P)).

To confirm suppression of the PBP1a-only growth defect by the MraY variant, *Pa*mraY(WT) or *Pa*mraY(T23P) was expressed from a plasmid under the control of an isopropyl-β-D-1-thiogalactopyranoside

(IPTG)-inducible promoter in a wild-type *P. aeruginosa* (strain PAO1) or a Δ*ponB* Δ*lpoA* background (Fig. 1b). Overexpression of *[Pa]mraY*(WT) in the wild-type strain neither appreciably affected growth on either LB or VBMM, nor did it rescue the lethal phenotype of the Δ*ponB* Δ*lpoA* mutant on VBMM (Fig. 1b and Extended Data Fig. 1). Consistent with the results of the genetic selection, expression of *[Pa]mraY*(T23P) reduced the frequency of lysing cells observed in permissive conditions (Extended Data Fig. 1) and restored growth of the PBP1a-only mutant on VBMM (Fig. 1b and Extended Data Fig. 2). Notably, in addition to rescuing growth of the mutant on VBMM, overexpression of *[Pa]mraY*(T23P) caused a mild growth defect in both wild-type and Δ*ponB* Δ*lpoA* backgrounds when cells were grown on rich medium (Fig. 1b and Extended Data Fig. 2) (see below). Substitution of the catalytic residue D267 with Ala in the active site of *[Pa]*MraY(T23P) eliminated the toxicity of the variant when it was overproduced in wild-type cells on LB and greatly reduced the suppression activity in Δ*ponB* Δ*lpoA* cells on VBMM (Extended Data Fig. 2). Furthermore, functional VSVG-tagged derivatives of *[Pa]*MraY(WT) and *[Pa]*MraY(T23P) were found to accumulate to similar levels in cells by immunoblot analysis (Extended Data Fig. 3). Thus, the suppression activity of *[Pa]*MraY(T23P) is not due to increased accumulation of the enzyme. Rather, the results suggest that the T23P change alters MraY activity to promote the growth of the aPBP-deficient strain on VBMM and impair growth of both mutant and wild-type strains on LB when it is overexpressed.

*E. coli* also encodes aPBPs, *[Ec]*PBP1a and *[Ec]*PBP1b, controlled by OM lipoprotein activators *[Ec]*LpoA and *[Ec]*LpoB, respectively (Fig. 1c)[14,16]. We previously described an *E. coli* strain lacking *[Ec]*PBP1a and *[Ec]*LpoB that relies on an LpoB-bypass variant of *[Ec]*PBP1b [*[Ec]*PBP1b(E313D)] as its only aPBP (Fig. 1c)[17]. Similar to the *P. aeruginosa* Δ*ponB* Δ*lpoA* strain, this *E. coli* mutant has a conditional growth defect caused by a deficit in aPBP activity. It grows on LB without added NaCl (LBNS) but is inviable on LB with 1% NaCl. Overproduction of *E. coli* MraY(T23P) [*[Ec]*MraY(T23P)] but not wild-type *[Ec]*MraY suppressed the growth defect of this aPBP-deficient *E. coli* strain on LB with 1% NaCl (Fig. 1d). Therefore, an MraY(T23P) variant suppresses an aPBP defect in two distantly related gram-negative bacteria, suggesting that its properties are conserved.

## MraY(T23P) is activated and increases lipid II accumulation

We reasoned that MraY(T23P) might overcome the aPBP deficiency in mutants of *P. aeruginosa* and *E. coli* by increasing the cellular concentration of the aPBP substrate lipid II. Accordingly, *[Pa]*MraY(T23P) production in *P. aeruginosa* promoted better growth than *[Pa]*MraY(WT) on media containing carbenicillin, a beta-lactam antibiotic that places excess demand on PG precursor production[18] (Supplementary Fig. 1). We therefore directly measured the concentration of lipid II in *P. aeruginosa* and *E. coli* cells overproducing MraY(WT) or MraY(T23P) (Fig. 2). In both the wild-type and aPBP-deficient mutant backgrounds, MraY(WT) overproduction led to an approximately twofold increase in lipid II levels relative to an empty vector control (Fig. 2c,e). The increase was another twofold higher for cells overproducing MraY(T23P) (Fig. 2c,e). We observed similar trends when monitoring lipid I levels, but the increase in lipid I levels in cells producing MraY(T23P) relative to MraY(WT) was not nearly as pronounced compared with the change in lipid II levels (Extended Data Fig. 4). These results suggest that the altered MraY enzyme is more active than the wild type and that the ability to promote the accumulation of higher lipid II levels indeed underlies the suppression of aPBP defects.

To assess the effect of the T23P substitution on MraY activity directly, FLAG-tagged derivatives of *[Pa]*MraY(WT) and *[Pa]*MraY(T23P) were heterologously expressed in *E. coli* and affinity purified for biochemical assays. The reaction was followed by monitoring the production of uridine derived from alkaline phosphatase treatment of the UMP product (Fig. 2f). Using this assay, the *[Pa]*MraY(T23P) variant was found to be significantly more active than *[Pa]*MraY(WT). At the conclusion of

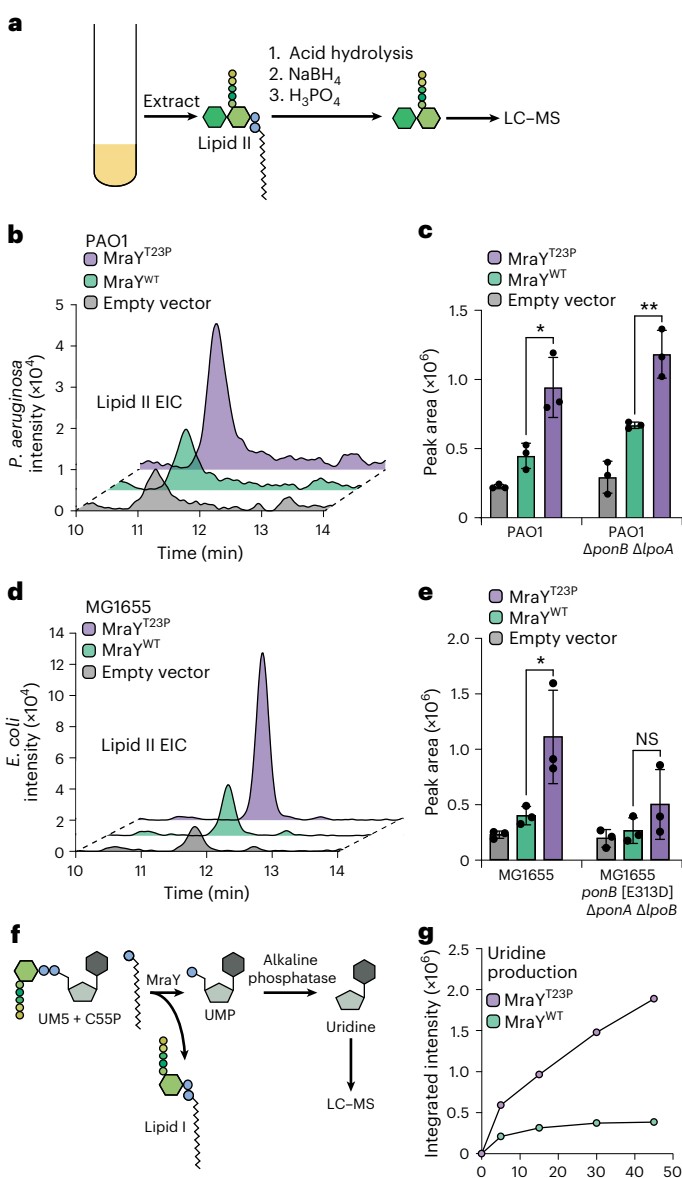

**Fig. 2 | Cells expressing MraY(T23P) accumulate lipid II. a**, Schematic representation of the method used to isolate and analyse lipid II from bacterial cells. **b–e**, Representative extracted ion chromatograms of lipid II (EIC) and quantification of EICs for *P. aeruginosa* (**b,c**) or *E. coli* (**d,e**) strains expressing the indicated MraY variant. Three independent replicates of the extractions were performed and lipid II levels quantified using the area of the peak from the extracted ion chromatogram using the Agilent software. Dots represent the values obtained for the biological replicates; bars and error bars indicate mean ± s.d. For PAO1: MraY[T23P] vs MraY[WT] in PAO1 *P = 0.0219, PAO1 Δ*ponB* Δ*lpoA* **P = 0.007; for MG1655: MraY[T23P] vs MraY[WT] in MG1655 *P = 0.0456, MG1655 Δ*ponA* Δ*lpoB* *ponB*[E313D], NS (not significant), (unpaired two-tailed *t*-test) but the trends for the individual samples were consistent with the other experiments with *[Pa]*MraY[T23P] production promoting the highest levels of lipid II and empty vector the least. **f**, Schematic representation of the MraY enzyme assay. **g**, Representative time course showing the production of uridine in assays containing purified MraY or MraY[T23P] as indicated. The assay was repeated at least twice with two independent preparations of protein. NaBH₄, sodium borohydride; H₃PO₄, phosphoric acid.

the time course, approximately five times more uridine was detected in reactions containing *[Pa]*MraY(T23P) than those with *[Pa]*MraY(WT) (Fig. 2g). We conclude that the T23P substitution generates a hyperactive MraY, leading to elevated lipid II production in cells.

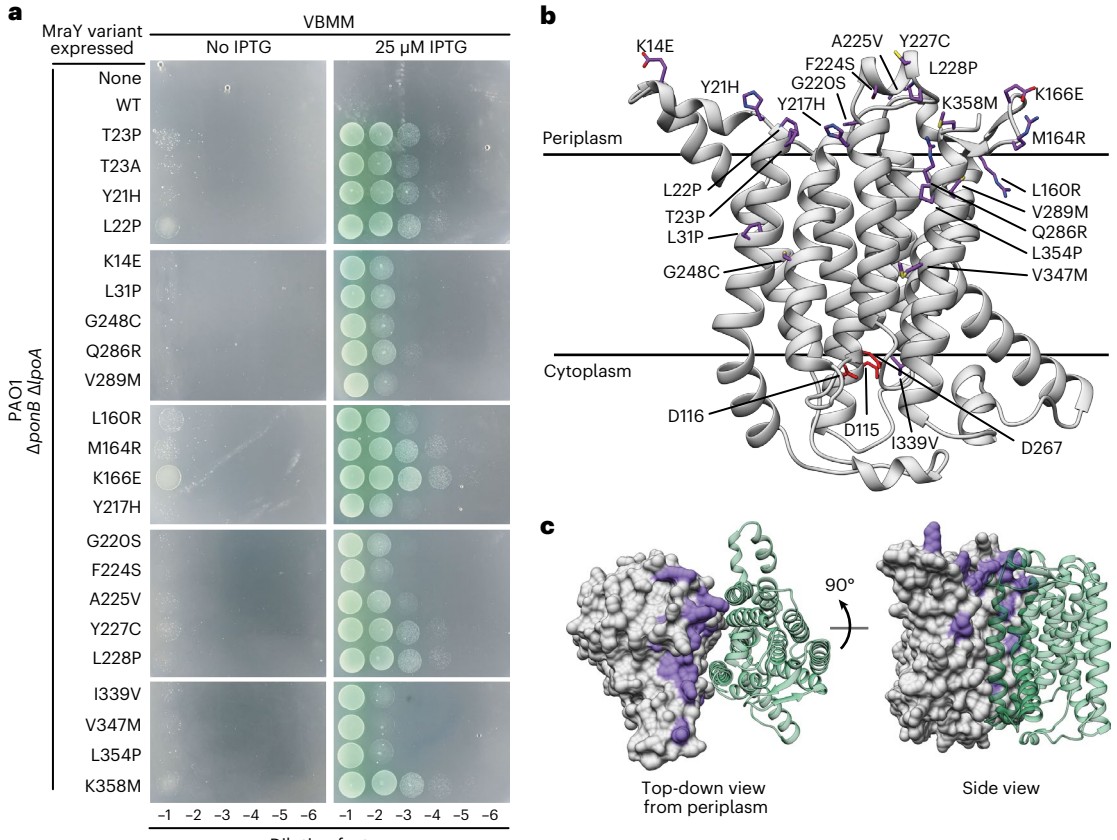

**Fig. 3 | Amino acid substitutions in hyperactive MraY variants localize to the extracytoplasmic surface of the dimer interface. a**, Tenfold serial dilutions of *P. aeruginosa* Δ*ponB* Δ*lpoA* cells harbouring expression plasmids producing the indicated MraY variant were plated on VBMM with or without IPTG to induce the MraY variants as indicated. **b**, Structural model of *P. aeruginosa* MraY created using AlphaFold[24] in cartoon viewed from the plane of the membrane. Residues altered in hyperactive variants tested in **a** are shown in stick representation (purple), while those residues previously implicated in catalysis are shown in red. **c**, Structural model of the MraY dimer created using AlphaFold[24]. Surface representation of one protomer is shown in grey, with the residues altered in hyperactive variants coloured in purple. The other protomer is shown in cartoon representation (green) for simplicity. Left: periplasmic view of the dimer. Right: view from the plane of the membrane.

## Misregulated MraY disrupts O-antigen synthesis

We wondered whether excess lipid II production and the resulting sequestration of C55P in this building block indirectly impede the synthesis of other surface glycans such as O-Ag that are built on the lipid carrier. A clue that this was the case came from the growth defect on LB medium of the wild-type strain, caused by overproduction of [Pa]MraY(T23P) but not [Pa]MraY(WT) (Fig. 1b and Extended Data Fig. 2). Notably, this *P. aeruginosa* strain produces R2-pyocin, a lethal phage tail-like bacteriocin that uses a receptor located within the LPS core to engage target cells[19,20]. *P. aeruginosa* is resistant to killing by its own R2-pyocin because it decorates its LPS with O-Ag that masks the R2-pyocin receptor. Defects in the O-Ag synthesis pathway therefore result in susceptibility to R2-pyocin self-killing[21]. The connection between O-Ag and R2-pyocin activity suggested to us that the growth phenotype induced by [Pa]MraY(T23P) overproduction on LB medium may be caused by a decrease in O-Ag production and increased R2-pyocin self-intoxication. To test this possibility, we examined the effect of [Pa]MraY(T23P) overproduction in a strain deleted for the R2-pyocin gene cluster (*PA0615–PA0628*). Unlike wild-type cells, the mutant incapable of making R2-pyocin was largely unaffected by the overproduction of [Pa]MraY(T23P) (Extended Data Fig. 5a), indicating that the growth defect caused by the altered enzyme was largely due to R2-pyocin killing. This result suggested that O-Ag synthesis is reduced when lipid II synthesis is hyperactivated in cells producing [Pa]MraY(T23P). Analysis of the LPS produced by these cells confirmed

that they indeed have reduced levels of O-Ag. They made approximately 30% less O-Ag compared with cells expressing [Pa]MraY(WT) (Extended Data Fig. 5b,c). In addition, overproduction of the WbpL initiator transferase for O-Ag synthesis was found to reduce the ability of [Pa]MraY(T23P) to suppress the growth defect of the PBP1a-only strain (Extended Data Fig. 5d). These results suggest that [Pa]MraY(T23P) may be insensitive to a regulatory mechanism limiting the steady-state accumulation of lipid-linked PG precursors to prevent the impairment of competing pathways utilizing the C55P carrier.

## A potential regulatory site on MraY

MraY is a polytopic membrane protein with ten transmembrane helices[22]. The structure of the enzyme from *Aquifex aeolicus* revealed that it forms a dimer with most of the monomer–monomer contacts made between the N- and C-terminal helices[22]. Notably, the T23 residue lies near the dimer interface on the extracytoplasmic side of MraY. We therefore wondered whether other substitutions in this area might also activate the enzyme. To test this possibility, a mutagenized copy of [Pa]*mraY* under the control of an IPTG-inducible promoter was transformed into the Δ*ponB* Δ*lpoA P. aeruginosa* strain. The resulting transformants were then selected on VBMM in the presence of IPTG to identify MraY variants that rescue the aPBP deficiency. Twenty-one suppressing clones were isolated that each contained a single point mutation in the plasmid-borne copy of *mraY* (Fig. 3a). The positions of these substitutions were mapped onto a model of the [Pa]MraY structure generated

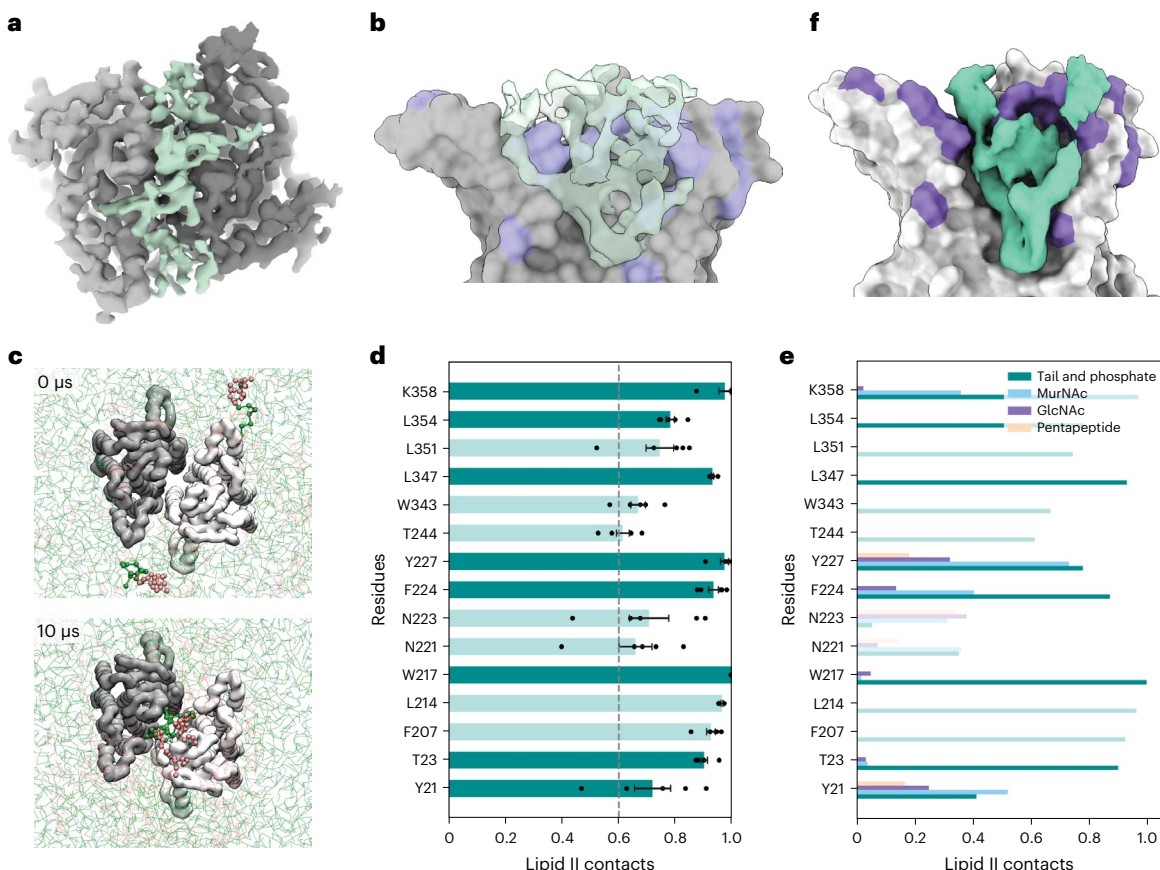

**Fig. 4 | Identification of a potential lipid II binding site in MraY. a**, Periplasmic view of the cryo-EM structure of the $^{Ec}$MraY(T23P) dimer within the YES complex shown in surface representation, with the unmodelled electron density shown (green). **b**, As in **a**, but membrane view of the electron density within the dimer interface of the $^{Ec}$MraY dimer with the foreground MraY removed. **c**, Top view of the MraY dimer in a mixed CG membrane. Two lipid II molecules (highlighted as green and pink spheres) freely enter the MraY cavity during unbiased MD simulations. In 8/9 repeats, 2 or 3 lipid I or II molecules bind the cavity. In the last repeat, one lipid II and one C55P molecule bind. **d**, Lipid II contacts with MraY residues that interact with lipid II for over 60% of atomistic MD simulations, where the dashed line indicates the 60% cutoff. Dots represent the values obtained for the independent replicates; bars and error bars indicate mean ± s.e. from 5 repeats. Darker green bars represent residues altered in hyperactive variants. **e**, Lipid II contacts with MraY residues by part of lipid II that is interacting (tail and phosphate, MurNAc, GlcNAc or pentapeptide). Residues shown are the same as those in **d**. Darker bars represent residues altered in hyperactive variants. **f**, Average density of lipid II molecules (green) from atomistic MD simulations of MraY (grey) bound to lipid II. Shown as inside view of dimer interface, where only one monomer of MraY is shown and residues altered in hyperactive variants are coloured in purple.

using AlphaFold[23,24]. Strikingly, all changes were located proximal to the dimer interface, with a majority positioned on the extracytoplasmic side of the protein far from the active site, which is located on the cytoplasmic side of the enzyme (Fig. 3b,c and Supplementary Table 1). Overall, our genetic and biochemical results implicate the extracytoplasmic region of MraY near the dimer interface as a potential regulatory site for the enzyme.

**A potential binding site for lipid II within the MraY dimer**

Both the *A. aeolicus* and *Enterocloster boltae* MraY structures revealed the presence of a cavity located at the dimer interface that is lined by hydrophobic residues[22,25]. This hydrophobic cavity is a conserved feature of the enzyme (Extended Data Fig. 6) and it was suggested that the electron density within it could accommodate one or more lipid molecules. Although it has been speculated to be C55P[22], the identity of the lipid has remained unclear. In addition, a recent study identified lipid molecules co-purifying with MraY, including C55P, lipid I and lipid II[26]. Thus, MraY probably binds a lipid molecule within the dimer interface near residues we have implicated in controlling the activity of the enzyme.

Clues to the potential identity of the lipid bound at the MraY dimer interface came from structural analysis of $^{Ec}$MraY in complex with a

phage-encoded inhibitor (protein E) and the *E. coli* chaperone SlyD (the YES complex)[27]. The cryogenic electron microscopy (cryo-EM) structure of the YES complex containing wild-type $^{Ec}$MraY was recently reported[27], and this methodology was used to obtain the structure of $^{Ec}$MraY(T23P) within the same complex (Supplementary Fig. 2 and Supplementary Table 2). In both cases, electron density was observed at the MraY dimer interface. Focused refinement of MraY alone in the $^{Ec}$MraY(T23P) complex substantially improved the potential lipid density at the MraY dimer interface (Fig. 4a,b). As in previous *A. aeolicus* and *E. boltae* MraY structures, this electron density fills the hydrophobic cavity found at the MraY dimer interface. However, we uniquely observed this electron density extending into the periplasmic space above the MraY molecules where the environment is more hydrophilic (Fig. 4a,b). Although structural refinement could not conclusively identify the lipid within the dimer, the size of the electron density extending into the periplasmic space is consistent with a large head-group such as the disaccharide-pentapeptide found on lipid II.

To assess whether a lipid II molecule could enter the hydrophobic cavity of the MraY dimer, we performed molecular dynamics (MD) simulations using the structure of the *E. coli* MraY dimer from the YES complex (PDB 8G01)[27] embedded in a lipid bilayer containing C55P, C55PP, lipid I or lipid II, with hydrophilic head-groups oriented towards what

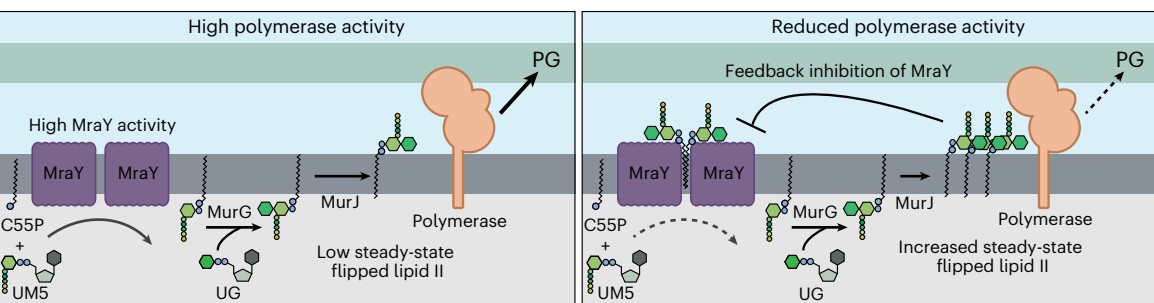

**Fig. 5 | Model for feedback regulation of MraY by flipped lipid II.** Shown are schematics summarizing the model for MraY regulation. Left: when PG polymerase activity is high, flipped lipid II is consumed at a rate proportional to its production such that steady-state levels of the precursor remain low and MraY activity is unimpeded. Right: when PG polymerase activity is reduced due to changes in growth conditions or other perturbations, lipid II will be produced faster than it is consumed, resulting in the accumulation of elevated levels of flipped lipid II. Higher levels of the precursor promote its binding to MraY dimers, reducing their activity to bring lipid II supply back in balance with demand by the polymerases. See text for details. UG, UDP-GlcNAc.

would be the periplasmic side of the membrane. Using coarse-grained MD simulations, we observed that in almost all runs, lipid I and lipid II molecules spontaneously entered the central cavity, where typically two molecules would occupy the cavity (Fig. 4c and Supplementary Video 1). Bilayer phospholipids never entered the cavity, while C55P and C55PP would occasionally enter the cavity at ~4% and 10% of the simulation time (Supplementary Video 2). Lipid I and lipid II would remain stably bound for at least several μs (Supplementary Video 1), reflected by the rate constants for complex dissociation ($k_{off}$) values of 0.218 μs$^{-1}$ and 0.206 μs$^{-1}$ for lipid I and lipid II, respectively. In similar experiments where lipid I and lipid II were omitted, a single cardiolipin (CDL) or C55P molecule would enter the cavity (Supplementary Video 2), although in the majority of simulations, no lipid entered the cavity. These were much shorter-lasting interactions, with $k_{off}$ values of 3.976 μs$^{-1}$ for C55P and 1.297 μs$^{-1}$ for CDL. Although lipid I and lipid II had similar $k_{off}$ values in these simulations, lipid I was not found in the periplasmic leaflet of the inner membrane. Therefore, the simulations with lipid I are not likely to reflect a physiologically relevant binding event. Instead, lipid II is the best candidate for the native ligand due to its strong and long-lasting interaction. Notably, the bound lipid II molecules in the simulations make extensive contacts with the MraY dimer, with many residues contacting the bound lipids for nearly 100% of the MD simulations (Fig. 4d and Extended Data Fig. 7). These residues include several that were identified in the mutational analysis as being hyperactive (Fig. 3a,b). To investigate the interaction in more detail, a pose of the *E. coli* MraY dimer with two bound lipid II molecules was converted to an atomistic description for further MD analysis. The data show that the lipid II molecules are stable in the central cavity, with the isoprenyl chains adopting a curved orientation. The result predicts contacts between the MurNAc sugar and MraY that include several residues where substitutions were identified in our screen (Y21, T23, W217, F224, Y227 and K358) (Fig. 4e,f). Together, these data indicate that C55P-linked lipids can spontaneously enter a previously empty MraY dimer interface cavity and that externalized lipid II is likely to be the ligand bound in the potential regulatory site identified in the genetic and biochemical analyses.

## Discussion

Bacterial surfaces contain multiple types of glycan and other polymers that are required for cellular integrity and/or barrier function. Although most of the proteins involved in the synthesis of major surface components are known, how the biogenesis of these molecules is regulated to efficiently distribute shared precursors, such as the C55P lipid carrier, among competing synthesis pathways remains poorly understood. In this report, we uncover a mechanism governing the activity of MraY, the essential enzyme catalysing the first membrane-associated step in the PG synthesis pathway in which C55P is consumed to form lipid-linked

PG precursors. This regulation is likely to play an important role in the efficient distribution of C55P among glycan biogenesis pathways that utilize the limiting carrier.

The first clue that MraY is regulated came from the discovery that an *mraY*(*T23P*) mutant can suppress an aPBP deficiency in both *P. aeruginosa* and *E. coli*. The aPBP-deficient strains encode a single aPBP lacking its required activator. Previous work with these strains suggests that their conditionally lethal growth phenotypes are caused by poor PG synthesis efficiency resulting from the synthase having a reduced affinity for lipid II in the absence of its activator[15]. Accordingly, we infer that MraY(T23P) suppresses this problem by raising the steady-state level of lipid II to overcome the substrate binding limitations of the unactivated aPBP. The ability of the altered MraY to increase lipid II levels indicates a role for the enzyme in regulating the maximum level of lipid II in cells. We propose that this control is mediated via feedback inhibition of MraY by externalized lipid II (Fig. 5).

In support of the feedback inhibition model, the biochemical results with purified enzymes indicate that the observed regulation is intrinsic to MraY and does not require additional proteins. The MraY(T23P) variant, which is apparently less sensitive to regulatory control, showed much greater activity in vitro than the wild-type enzyme. At first glance, this result may seem incompatible with the proposed feedback control given that the product of the reaction is lipid I with its head-group in the cytoplasm, not externalized lipid II. However, because the reactions are performed in detergent, the lipid I formed in the reaction is probably capable of reorienting in the micelles to mimic a periplasmic orientation. Although externalized lipid I is not observed in vivo, the MD simulations predict that both flipped lipid I and lipid II are capable of binding at the MraY dimer interface. It is therefore reasonable to interpret the biochemical results in the context of a feedback inhibition model with MraY(WT) activity leveling off early in the time course due to feedback control. By contrast, we infer that MraY(T23P), with its substitution in the proposed binding site for flipped lipid II, is insensitive to feedback control and therefore displays robust activity in the assay. Another factor that is likely to contribute to the biochemical results is the co-purification of lipid II with the purified enzymes, which according to the model would be expected to further reduce the activity of MraY(WT) relative to MraY(T23P). Importantly, the activity for the wild-type enzyme was already so low that it was not possible to directly test for feedback inhibition via the addition of purified lipid II to the enzyme. Nevertheless, based on the logic above and the totality of the results presented, feedback inhibition of MraY by flipped lipid II provides one of the most straightforward explanations for our findings.

Although additional experiments are required to further investigate the possible feedback regulation of MraY, it is a compelling model because it suggests a mechanism by which cells can balance the supply

of flipped lipid II precursor with the activity of the PG synthases that use it to build the cell wall (Fig. 5). We propose that when PG synthases are highly active, the steady-state level of lipid II remains low such that MraY is functioning near its maximum activity to continue supplying lipid-linked PG precursors (Fig. 5, left panel). However, when the supply of lipid II exceeds the capacity of the PG synthases to use it, either transiently or due to a change in growth conditions, the steady-state level of lipid II will rise such that it begins binding MraY dimers to inhibit their activity and reduce flux through the lipid stages of PG precursor production until supply more closely matches demand (Fig. 5, right panel). Such feedback control would prevent excess C55P from being sequestered in PG precursors when they are not needed, making more of the lipid carrier available to other glycan synthesis pathways for their efficient operation. Accordingly, *P. aeruginosa* cells with an activated MraY variant, which is presumably less sensitive to feedback control, display reduced ability to make O-Ag, rendering them susceptible to self-intoxication by their encoded pyocins (Extended Data Fig. 5b,c).

The location of the amino acid substitutions in MraY that suppress aPBP defects combined with the structural and MD analysis suggest a mechanism by which the enzyme may be regulated by lipid II binding. Many of the MraY substitutions that overcome the PG synthesis defects of the PBP1a-only strain localize to the extracytoplasmic surface of the protein distal to, and on the other side of the membrane from, the active site. These changes flank the opening of a deep hydrophobic pocket at the MraY dimer interface (Extended Data Fig. 6). In the cryo-EM structure of MraY within the YES complex[27], we observe an MraY dimer with electron density at this interface as observed in previous X-ray crystal structures[22,25]. However, in our structure, this density not only fills the pocket but also extends into the extracytoplasmic opening. This density in the extracytoplasmic space is large enough to correspond to a head-group of flipped lipid II. Accordingly, MD simulations indicate the capacity of MraY dimers to bind two molecules of flipped lipid II, with contacts between the protein and the MurNAc sugar that probably provide specificity for externalized lipid II binding over C55PP or C55P. Notably, the head-groups of the lipid II binding substrates remain relatively flexible in the simulations (Supplementary Video 1 and Extended Data Fig. 8), which probably accounts for our inability to further refine the structure of the bound molecules by cryo-EM.

The MD simulations predict conformational changes in the MraY dimers associated with lipid II binding that increase the distance between the 6th transmembrane helix (TM6) of each monomer in the dimeric structure and alter the position of the 9th transmembrane helix (TM9) (Extended Data Fig. 9a–d). Similarly, the distance between a periplasmic helix (residues 221–228) from each monomer is also increased (Extended Data Fig. 9c–f). These changes are reminiscent of the conformational difference between MraY in the YES complex and the free MraY structure from *A. aeolicus*[22]. When the structures are aligned on one monomer, the second monomer in the YES complex[27] is tilted relative to its partner in the *A. aeolicus* dimer[22], resulting in the opening of the periplasmic cavity and tightening of the interface at the cytoplasmic side of the enzyme where the active site is located (Supplementary Fig. 3). Because MraY in the YES complex is inhibited by the phage lysis protein, this opened conformation probably represents the inhibited state. The similarities between the conformational changes in MraY observed in the YES complex and upon lipid II binding in the MD analysis indicate that it is feasible for lipid II binding on the periplasmic side of the enzyme to be communicated to the active site via an alteration of the dimer interface. Accordingly, an increased mobility of TM9 on the cytoplasmic face is also observed in the MD analysis when lipid II is bound (Extended Data Fig. 9b). How the T23P and other changes presumably activate MraY by reducing the sensitivity of the enzyme to inhibition by lipid II is not yet clear. However, electron density corresponding to the lipid is still observed at the dimer interface between MraY(T23P) protomers in the variant YES complex. Although this result may be affected by the enzyme being stuck in an inhibited state by the phage inhibitor, it suggests that T23P and other changes in MraY may affect the conformational response of the enzyme to lipid II binding rather than the binding event itself. Consistent with this possibility, tyrosine at position 21 has an altered conformation in the MraY(T23P) structure in which its hydroxyl group forms a hydrogen bond network with Y227 and K358 on the opposing monomer (Extended Data Fig. 10). Substitutions within these residues were also identified in the screen for hyperactive MraY enzymes, and Y227 is in the periplasmic helix that was found to be altered in the MD analysis upon lipid II binding. Thus, alterations affecting interactions in this region may be responsible for the regulation of MraY activity and its potential modulation by lipid II binding.

MraY belongs to the polyprenyl-phosphate *N*-acetylhexosamine 1-phosphate transferase (PNPT) superfamily of proteins that are found in all domains of life. The superfamily includes enzymes that initiate the lipid-linked stages of many glycan polymers including O-antigens, capsules and teichoic acids in bacteria. A well-studied example outside of bacteria is the GlcNAc-1-P-transferase (GPT) that catalyses the first step of N-linked protein glycosylation in eukaryotes by conjugating GlcNAc to the lipid carrier dolichol phosphate (DolP) to form Dol-PP-GlcNAc[11]. In each synthesis pathway, the final lipid-linked precursor for each glycan is built on a lipid carrier that must be shared with other pathways. It would therefore not be surprising if externalized versions of the final lipid-linked precursors of many different glycan biogenesis pathways exerted feedback control on the PNPT superfamily member that initiates precursor synthesis. Such a broad utilization of this feedback regulation would provide a mechanism to efficiently distribute limiting lipid carrier molecules between competing glycan synthesis pathways in cells by matching precursor supply with utilization.

In summary, we provide evidence that the essential and broadly conserved MraY step in PG synthesis is subject to a previously unknown regulatory mechanism. Mutational and structural evidence identified the likely regulatory site on the enzyme. Importantly, this site is accessible by small molecules from the extracytoplasmic side of the membrane unlike the active site, which is in the cytoplasm. This regulatory site therefore represents an attractive new target for the development of small molecule inhibitors of MraY for potential use as antibiotics.

# Methods

## Plasmid construction

**pNG93 [P$_{lacUV5}$::*$^{Pa}$mraY*(*PA4415*)] is a pPSV38 derivative.** pPSV38 was digested with EcoRI/XmaI to generate the plasmid backbone. *P. aeruginosa mraY* (PA4415; M1-R360) was amplified from PAO1 genomic (g) DNA with oNG338/oNG339 to introduce a synthetic ribosome binding site (RBS) (5′-GAGGAGGATACAT-3′). After digestion with EcoRI/XmaI, the PCR product was ligated into pPSV38 to generate pNG93. The final construct was sequence verified using primers 556 and 557. The *mraY* gene in this and all related constructs below is inducible with IPTG.

**pNG102 [P$_{lacUV5}$::*$^{Pa}$mraY*(*T23P*)] is a pPSV38 derivative.** pPSV38 was digested with EcoRI/XmaI to generate the plasmid backbone. *P. aeruginosa mraY*(T23P) was amplified from PA760 via colony PCR with oNG338/oNG339 to introduce a synthetic RBS (5′-GAGGAGGATACAT-3′). After digestion with EcoRI/XmaI, the PCR product was ligated into pPSV38 to generate pNG93. The final construct was sequence verified using primers 556 and 557.

**pLSM116 [P$_{T7}$::*H-SUMO-FLAG-$^{Pa}$mraY*] is a pCOLADuet derivative.** The gene encoding full-length *P. aeruginosa mraY* was amplified from pNG93 using the primers oLSM302 and oLSM303. Using pCOLADuet as a template, the backbone was amplified using oLSM301 and oLSM304. The fragments were joined using Gibson assembly and sequence verified using primers 34 and 2325.

**pLSM117 [$P_{T7}$::*H-SUMO-FLAG-$^{Pa}$mraY(T23P)*]] is a pCOLADuet derivative.** The gene encoding full-length *P. aeruginosa mraY(T23P)* was amplified from pNG102 using the primers oLSM302 and oLSM303. Using pCOLADuet as a template, the backbone was amplified using oLSM301 and oLSM304. The fragments were joined using Gibson assembly and sequence verified using primers 34 and 2325.

**pLSM124 [$P_{lacUV5}$::$^{Ec}$*mraY*]] is a pPSV38 derivative.** The gene encoding full-length *E. coli mraY* was amplified from MG1655 gDNA using primers oLSM312 and oLSM313. Using pNG93 as a template, the backbone was amplified using oLSM311 and oLSM314. The fragments were joined using Gibson assembly. The final construct was sequence verified using primers 556 and 557.

**pLSM125 [$P_{lacUV5}$::$^{Ec}$*mraY(T23P)*]] is a pPSV38 derivative.** Using pLSM124 as a template, T23 was mutated to P using site-directed mutagenesis (QuikChange Lightning, Agilent) employing the primers oLSM315 and oLSM316. The final construct was sequence verified using primers 556 and 557.

**pLSM141 [$P_{lac}$::$^{Pa}$*mraY*]] is a pRY47 derivative.** The gene encoding full-length *P. aeruginosa mraY* was amplified from pNG93 using primers oLSM372 and oLSM373. Using pRY47 as a template, the backbone was amplified using oLSM374 and oLSM368. The fragments were joined using Gibson assembly. The final construct was sequence verified using primers 556 and 48.

**pLSM142 [$P_{lac}$::$^{Pa}$*mraY(T23P)*]] is a pRY47 derivative.** The gene encoding full-length *P. aeruginosa mraY(T23P)* was amplified from pNG102 using primers oLSM372 and oLSM373. Using pRY47 as a template, the backbone was amplified using oLSM374 and oLSM368. The fragments were ligated using Gibson assembly. The final construct was sequence verified using primers 556 and 48.

**pLSM143 [$P_{lac}$::$^{Ec}$*mraY*]] is a pRY47 derivative.** The gene encoding full-length *E. coli mraY* was amplified from pLSM124 using primers oLSM375 and oLSM376. Using pRY47 as a template, the backbone was amplified using oLSM377 and oLSM368. The fragments were ligated using Gibson assembly. The final construct was sequence verified using primers 556 and 48.

**pLSM144 [$P_{lac}$::$^{Ec}$*mraY(T23P)*]] is a pRY47 derivative.** The gene encoding full-length *E. coli mraY(T23P)* was amplified from pLSM125 using primers oLSM375 and oLSM376. Using pRY47 as a template, the backbone was amplified using oLSM377 and oLSM368. The fragments were ligated using Gibson assembly. The final construct was sequence verified using primers 556 and 48.

**pLSM176 [$P_{lacUV5}$::$^{Pa}$*mraY-GS-VSVG*]] is a pPSV38 derivative.** The gene encoding full-length *P. aeruginosa mraY* was amplified from pNG93 using primers oLSM317 and oLSM405. Using pNG93 as a template, the backbone was amplified using oLSM404 and oLSM318. The fragments were ligated using Gibson assembly. The final construct was sequence verified using primers 556 and 557.

**pLSM177 [$P_{lacUV5}$::$^{Pa}$*mraY(T23P)-GS-VSVG*]] is a pPSV38 derivative.** The gene encoding full-length *P. aeruginosa mraY(T23P)* was amplified from pNG102 using primers oLSM317 and oLSM405. Using pNG102 as a template, the backbone was amplified using oLSM404 and oLSM318. The fragments were ligated using Gibson assembly. The final construct was sequence verified using primers 556 and 557.

**pLSM196 [$P_{lacUV5}$::$^{Pa}$*mraY(D267A)*]] is a pPSV38 derivative.** Site-directed mutagenesis (QuikChange Lightning, Agilent) of pNG93 was performed to make the D267A change using oLSM460 and oLSM461.

**pLSM197 [$P_{lacUV5}$::$^{Pa}$*mraY(T23P/D267A)*]] is a pPSV38 derivative.** Site-directed mutagenesis (QuikChange Lightning, Agilent) of pNG102 was performed to make the D267A change using oLSM460 and oLSM461.

**pLSM195 [$P_{lacUV5}$::$^{Pa}$*wbpL*]] is a pPSV38 derivative.** The gene encoding full-length *P. aeruginosa wbpL* (*PA3145*) was amplified from PAO1 gDNA using primers oLSM458 and oLSM459. Using SacI and XbaI, the amplified PCR product was digested and ligated into empty pPSV38. The final construct was sequence verified using primers 556 and 557.

## Materials

Unless otherwise indicated, all chemicals and reagents were purchased from Sigma-Aldrich. Restriction enzymes were purchased from New England Biolabs. Oligonucleotide primers were purchased from Integrated DNA Technologies.

## Bacterial strains, plasmids, oligonucleotide primers and culture conditions

*E. coli* strains were grown with shaking at 37 °C in LB (10 g l⁻¹ tryptone, 5 g l⁻¹ NaCl, 5 g l⁻¹ yeast extract), LBNS (10 g l⁻¹ tryptone, 5 g l⁻¹ yeast extract), TB (12 g l⁻¹ tryptone, 24 g l⁻¹ yeast extract, 0.4% v/v glycerol, 0.17 M $KH_2PO_4$, 0.72 M $K_2HPO_4$) or on LB or LBNS agar as indicated. MM119 was grown at 30 °C. *P. aeruginosa* strains are all derivatives of PAO1 and were grown with shaking at 37 °C in LB, LBNS, VBMM (3.42 g l⁻¹ trisodium citrate dihydrate, 2.0 g l⁻¹ citric acid, 10 g l⁻¹ $K_2HPO_4$, 3.5 g l⁻¹ $NaNH_4PO_4$·$4H_2O$, 1 mM $MgSO_4$, 0.1 mM $CaCl_2$) or on LB, LBNS or VBMM agar as indicated. The following concentrations of antibiotics were used to maintain plasmids: ampicillin (Amp), 50 µg ml⁻¹; chloramphenicol (Cam), 25 µg ml⁻¹; gentamicin (Gent), 15 µg ml⁻¹ (*E. coli*); Gent, 30 µg ml⁻¹ (*P. aeruginosa*). The primers, strains and plasmids used in this study are summarized in Supplementary Tables 3–5.

## Electroporation of *P. aeruginosa*

*P. aeruginosa* strains were made competent using previously described methods[28]. For electroporation, 100 ng of plasmid DNA was added to 40 µl of competent *P. aeruginosa* cells. Transformation was achieved using standard protocols and transformants were selected for using 30 µg ml⁻¹ Gent.

## Viability assays

Overnight cultures of PAO1, PA686 or PA760 derivatives containing vectors producing the indicated alleles of *mraY* expressed from an IPTG-inducible ($P_{lacUV5}$) plasmid were normalized to an optical density (OD)$_{600}$ of 2.4 before being serially diluted. Aliquots (5 µl) of the dilutions were spotted onto LB Gent agar or VBMM Gent agar, with or without IPTG. Plates were incubated at 30 °C for 24 h, at which point the plates were imaged. A similar protocol was adapted for MG1655 and MM119 derivatives containing vectors producing the indicated alleles of *mraY* from an IPTG-inducible ($P_{lac}$) plasmid.

## Immunoblotting

For analysis of protein levels from strains producing MraY-VSVG variants, an overnight culture of each of the strains was allowed to grow in LB containing 30 µg ml⁻¹ Gent at 37 °C. The following day, the cultures were diluted to an OD$_{600}$ of 0.01 and allowed to grow at 37 °C in LB containing 30 µg ml⁻¹ Gent. After 2 h, 1 mM IPTG was added and the cultures were allowed to grow for another 2.5 h. Cultures were normalized to an OD$_{600}$ of 1.0 and cells were collected by centrifugation at 5,000 × $g$ for 2 min. The cell pellet was resuspended in 200 µl of 2× Laemmli buffer and then centrifuged for 10 min at 21,000 × $g$. Samples were analysed by SDS–PAGE followed by immunoblotting. Protein was transferred from the SDS–PAGE gel to a nitrocellulose membrane using wet transfer (30 min at 100 V) in cold transfer buffer (192 mM glycine, 20% methanol, 25 mM Tris base). The membrane was blocked in 5% (w/v) skim milk powder

in Tris-buffered saline (10 mM Tris-HCl pH 7.5, 150 mM NaCl) containing 0.5% (v/v) Tween-20 (TBS-T) for 45 min at room temperature with gentle agitation. The α-VSVG antibody (V4888, Sigma) was added to the blocking buffer at a 1:5,000 dilution for 1 h. The membrane was washed three times in TBS-T for 5 min each before incubation for 1 h with secondary antibody (anti-rabbit IgG HRP, 1:5,000 dilution, 7074S, NEB) in TBS-T with 1% (w/v) skim milk powder. The membrane was then washed three times with TBS-T for 5 min each before developing using Clarity Max Western ECL substrate (1705062, BioRad) and imaged using a BioRad ChemiDoc XRS+.

## Microscopy and image acquisition

Cells were grown overnight in LB Gent at 37 °C. The following day, cultures were diluted 1:500 into LB Gent and cells were allowed to grow for 2 h at 37 °C before inducing expression of *mraY* using 1 mM IPTG. Cells were allowed to grow for a further 2.5 h before being immobilized on 1.5% LB agarose pads and covered with #1.5 coverslips. Phase-contrast microscopy images were obtained using a Nikon Eclipse Ci-L plus upright microscope fitted with a Nikon Digital Sight Fi3 6MP colour camera, a Plan Apo Lambda ×100/1.45 NA oil immersion objective lens and Nikon Elements F acquisition software.

## Error-prone PCR

Mutagenesis was adapted from ref. 29. Four independent mutant plasmid libraries were constructed by mutagenizing *mraY* in plasmid pNG93 ($P_{lacUV5}$::*mraY*) using *Taq* polymerase with Thermopol buffer (New England Biolabs, M0267L). The forward 5′-ACACTTTATGCTTCCGGCTC-3′ and reverse 5′-ACTGTTGGGAAGGGCGATCAAA-3′ primers were used to amplify *mraY* from pNG93. The resulting PCR products were purified using the Monarch PCR & DNA cleanup kit (NEB, T1030) and used as 'megaprimers' that were denatured and annealed to the original plasmid (pNG93) to amplify the vector backbone using Q5 High-Fidelity 2X master mix (NEB, M0492S). The reactions were then digested with DpnI to eliminate any remaining parental plasmid DNA. All four libraries were independently electroporated into NEB 10-beta electrocompetent cells (NEB, C3020K) and plated on LB agar supplemented with 15 μg ml⁻¹ Gent at 37 °C overnight.

Transformants were slurried in LB and the resuspended cells were normalized to an $OD_{600}$ of 10. Cells from 1 ml of resuspension were centrifuged and plasmid DNA was isolated from the cell suspension using the Monarch Plasmid DNA miniprep kit (T1010). All four libraries were independently transformed into electrocompetent PA686 cells, plated on LBNS agar supplemented with 30 μg ml⁻¹ Gent and grown overnight at 37 °C. The resulting transformant colonies from each of the libraries were slurried in LBNS supplemented with 30 μg ml⁻¹ Gent. Samples of each were normalized to $OD_{600}$ = 10 in LBNS + 10% (v/v) dimethylsulfoxide and stored at −80 °C. A sample from each library was then thawed, and serial dilutions were plated on VBMM with 30 μg ml⁻¹ Gent with or without IPTG (50 μM) and grown at 30 °C overnight. Individual colonies arising on the IPTG-supplemented plates from each library were selected and re-streaked on VBMM with or without IPTG. Those that displayed IPTG dependence were further isolated and the plasmids sent for sequencing. Clones identified to contain a single point mutation were further characterized. The mutated *mraY* genes were each amplified using Q5 High-fidelity polymerase (NEB) via colony PCR. The purified PCR product was digested with EcoRI and XmaI, and subsequently ligated into pPSV38 for validation of the suppression phenotype. All clones were sequence verified. MraY variants are listed in Supplementary Table 1.

## Lipid II extraction

Cultures of PAO1, PA686 and MG1655 were grown at 37 °C overnight and MM119 at 30 °C overnight. The next day, cultures were diluted to an $OD_{600}$ of 0.01 and allowed to grow for 2 h at the above specified temperatures, whereupon 1 mM IPTG was added to induce expression

of MraY. Cells were collected when the $OD_{600}$ reached ~0.5 and normalized to OD = 1 in a 1 ml volume. Pellets were collected by centrifugation at 21,000 × *g* and stored at −20 °C until needed. Cells were resuspended in 1 ml LB and added to a mixture of 2:1 methanol:chloroform (3.5 ml total) in borosilicate glass tubes (16 mm × 100 mm, Fisher Scientific, 1495935AA). Samples were vortexed for 1 min to form a single phase. Cell debris was collected by centrifugation for 10 min at 2,000 × *g* at 21 °C. The supernatant was transferred to a fresh borosilicate glass tube and 2 ml of chloroform was added. The supernatant was acidified using 0.1 N HCl to pH 1 as determined by pH indicator strips. The samples were vortexed for 1 min and centrifuged for 20 min at 2,000 × *g* at 21 °C to form a two-phase system. Using a glass pipette, as much of the aqueous upper layer was removed without disturbing the interface between the aqueous and organic phases, and 1 ml methanol was subsequently added to form a single liquid phase upon vortexing. Samples were transferred to 1.5 ml Eppendorf tubes by glass pipette and then dried by nitrogen stream at 40 °C. Dried samples were dissolved in 150 μl of a mixture of methanol and chloroform (2:1) by vortexing, then centrifuged at 21,000 × *g* for 1 min and dried by nitrogen stream at 40 °C. This was repeated with 40 μl organic mixture and finally, crude lipid extracts were dissolved in 10 μl dimethylsulfoxide by vortexing. Extracts were stored at −20 °C.

## Lipid II hydrolysis

Crude lipid II (LII) extracts (5 μl) were added to 5 μl of 0.2 M HCl, for a final concentration of 0.1 M HCl. Samples were boiled at 100 °C for 15 min and then cooled to 4 °C in a thermocycler. Sodium borate (10 μl, pH 9) was added, followed by 1 μl 0.5 M NaOH to neutralize the solution. Sodium borohydride (2 μl, 100 mg ml⁻¹) was added and the samples were allowed to incubate for 30 min at room temperature. Following the incubation, 2 μl of 20% phosphoric acid was added to quench the reaction, and the samples were mixed and immediately subjected to liquid chromatography mass spectrometry (LC–MS) analysis.

## LC–MS

High-resolution LC–MS traces of soluble LII hydrolysis products were obtained using the following protocol. Briefly, the hydrolysed samples were subjected to LC–MS analysis (ESI, positive mode). A Waters Symmetry Shield RP8 column (3.5 μm, 4.6 mm × 150 mm) was used to separate hydrolysis products using the following gradient (A, $H_2O$ + 0.1% formic acid; B, acetonitrile + 0.1% formic acid; 0.5 ml min⁻¹): 0% B for 5 min, followed by a linear gradient of 0–20% B over 40 min. Data were obtained on an Agilent 6546 LC-q-TOF mass spectrometer. Expected ion masses were extracted with a tolerance of 0.01 mass units.

## Purification of UDP-MurNAc pentapeptide

Accumulation of the precursor was performed as previously described[30] with the following modifications. *Bacillus cereus* ATCC 14579 was grown in LB-lennox medium at 37 °C until the $OD_{600}$ reached between 0.7 and 0.8, at which point 130 μg ml⁻¹ of chloramphenicol was added. After 15 min of incubation, 5 μg ml⁻¹ of vancomycin was added and the cells allowed to incubate for another 60 min at 37 °C with shaking. The culture was then cooled on ice and collected by centrifugation (4,000 × *g*, 20 min, SLC-6000 rotor, 4 °C). Cells were collected and stored at −20 °C until required.

Cells were resuspended in water (0.1 g wet weight ml⁻¹) and stirred into boiling water in a flask with stirring. Boiling was allowed to continue for another 15 min, at which point the flask was removed from heat and allowed to cool to room temperature with stirring. After ~20 min, the resuspension was cooled on ice and the debris was pelleted at 200,000 × *g* for 60 min at 4 °C. The supernatant was removed and lyophilised. The lyophilised material was resuspended in water, acidified to pH 3 using formic acid (1 ml l⁻¹ culture extracted), centrifuged to remove the precipitate and immediately subjected to reversed-phase high-pressure liquid chromatography (RP-HPLC).

UDP-MurNAc pentapeptide was isolated by RP-HPLC on a Synergi 4u Hydro-RP 80A column (250 mm × 10.0 mm). The column was eluted over a 30-min isocratic programme (A, $H_2O$ + 0.1 % formic acid; B, acetonitrile + 0.1% formic acid; 4 ml min$^{-1}$), 4% B for 30 min at room temperature. The elution was monitored by UV at 254 nm. UDP-MurNAc-pentapeptide eluted at ~20 min in a single peak, which was verified by mass spectrometry (1,194.35 Da). Peak fractions were collected and lyophilised. The final product was resuspended in water for downstream use.

### Expression and purification of $^{Pa}$MraY

For expression of *P. aeruginosa* MraY or MraY$^{T23P}$, *E. coli* expression strain LSM9 containing pAM174 and the expression plasmid (pLSM116 or pLSM117) was grown in 1 l TB supplemented with 2 mM MgCl$_2$, kanamycin and chloramphenicol at 37 °C with shaking until the OD$_{600}$ was 0.7. The cultures were cooled to 20 °C before inducing protein expression with 1 mM IPTG and 0.1% (w/v) arabinose. Cells were collected at 19 h post induction by centrifugation (6,000 × *g*, 15 min, 4 °C). To purify FLAG-MraY or FLAG-MraY$^{T23P}$, the cells were resuspended in lysis buffer B (50 mM HEPES pH 7.5, 150 mM NaCl, 20 mM MgCl$_2$, 0.5 mM dithiothreitol) and lysed by passage through a cell disruptor (Constant systems) at 25 kpsi twice. Membranes were collected by ultracentrifugation (100,000 × *g*, 1 h, 4 °C). The membrane pellets were resuspended in solubilization buffer B (20 mM HEPES pH 7.0, 0.5 M NaCl, 20% (v/v) glycerol and 1% (w/v) dodecyl 4-*O*-α-D-glucopyranosyl-β-D-glucopyranoside (DDM) (Thermo Fisher)), and rotated end over end for 1 h at 4 °C before ultracentrifugation (100,000 × *g*, 1 h, 4 °C). The supernatant was supplemented with 2 mM CaCl$_2$ and loaded onto a pre-equilibrated homemade M1 anti-FLAG antibody resin. The resin was washed with 25 column volumes (CVs) of wash buffer C (20 mM HEPES pH 7.0, 0.5 M NaCl, 20% (v/v) glycerol, 2 mM CaCl$_2$, 0.1% (w/v) DDM) and the bound protein was eluted from the column with 5 CVs of elution buffer (20 mM HEPES pH 7.0, 0.5 M NaCl, 20% (v/v) glycerol, 0.1% (w/v) DDM, 5 mM EDTA pH 8.0 and 0.2 mg ml$^{-1}$ FLAG peptide). Fractions containing the target protein were concentrated and the protein concentration was measured via the Bradford method. Proteins were aliquoted and stored at −80 °C until required.

### MraY translocase in vitro assay

The assay was performed at 37 °C in an assay buffer containing 20 mM HEPES pH 7.5, 500 mM NaCl, 20% (v/v) glycerol, 0.1% (w/v) DDM, 10 mM MgCl$_2$, 250 µM UDP-MurNAc pentapeptide and 1.1 mM C55P (Larodan). Protein was added to initiate the reaction at a final concentration of 1.7 µM. At the appropriate timepoint, the reaction was quenched by boiling for 3 min at 95 °C. Alkaline phosphatase (1.5 units) was added to the sample (NEB M0371L) and incubated at 25 °C for 1 h. The samples were heat quenched at 65 °C to stop the reaction and immediately loaded for analysis by LC−MS. The samples were monitored by UV 254 and by MS (ESI, positive mode). A Thermo Fisher Hypersil Gold aQ C18 (150 mm × 4.6 mm, 3 µm) HPLC column was used to separate the substrates and products using the following gradient programme (A, $H_2O$ + 0.1% formic acid; B, acetonitrile + 0.1% formic acid; 0.4 ml min$^{-1}$): 4% B for 20 min. Data were obtained on an Agilent 6546 LC-q-TOF mass spectrometer.

### Preparation of LPS and immunoblotting

To isolate LPS from the *P. aeruginosa* strains containing the indicated plasmids, overnight cultures of each of the strains were allowed to grow in LB containing 30 µg ml$^{-1}$ Gent at 37 °C. The next day, cultures were diluted to an OD$_{600}$ of 0.01 and allowed to grow at 37 °C in 25 ml LB containing 30 µg ml$^{-1}$ Gent. After 2 h, 1 mM IPTG was added and the cultures were allowed to grow for another 2 h until cultures reached mid-log. Culture (20 ml) was pelleted at 4,000 × *g* for 12 min, cells were resuspended in 1 ml LB and the OD$_{600}$ was measured. The cells were pelleted again at 12,000 × *g* for 2 min and resuspended in 1X LDS

buffer (Invitrogen, NP00008) + 4% β-mercaptoethanol (BME) to an OD$_{600}$ of 20. Samples were boiled at 95 °C for 10 min. Each sample was subjected to the NI protein assay (G Biosciences, 786-005) to determine the protein content in each sample. The lysates (50 µl) were then incubated at 55 °C with 1.25 µl proteinase K (NEB, P8107S). After 1 h of incubation, samples were boiled at 95 °C for 10 min and then frozen at −20 °C until required.

Volumes of lysates corresponding to 20 µg of protein were then run on a Criterion XT 4–12% Bis-Tris precast gel (BioRad, 3450124) in MES running buffer (50 mM MES, 50 mM Tris base, 1 mM EDTA, 0.1% (w/v) SDS) for 1 h and 45 min at 100 V constant. Glycan was transferred to nitrocellulose membranes as described above with the following differences: membranes were blocked for 1 h at room temperature in 1% (w/v) skim milk and were then incubated with anti-serotype O5 B-band at a 1:1,000 dilution overnight at 4 °C (gift from L. Burrows of Michael G. DeGroote Institute for Infectious Disease Research, Biochemistry and Biomedical Sciences, McMaster University). After three 15-ml TBS-T washes, membranes were incubated with anti-mouse HRP antibody (1:5,000, NEB 7076S) for 1 h at room temperature. Blots were developed as described above.

### MD simulations

For the coarse-grained MD, the structural model of the *E. coli* MraY dimer was aligned according to the plane of the membrane with memembed[31] and then converted to the Martini 3 force field using the martinize protocol[32]. Bonds of 500 kJ mol$^{-1}$ nm$^{-2}$ were applied between all protein backbone beads within 1 nm. Proteins were built into 13 nm × 13 nm membranes composed of 40% POPE and 10% each of POPG, CDL, lipid I, lipid II, C55P and C55PP using the insane protocol[33]. Alternatively, membranes were built with 60% POPE and 10% each of POPG, CDL, C55P and C55PP. Lipid I, lipid II, C55P and C55PP parameters were from ref. 27. Systems were solvated with Martini waters and Na$^+$ and Cl$^-$ ions to a neutral charge and 0.0375 M. Systems were minimized using the steepest descents method, followed by 1 ns equilibration using 5 fs time steps, then by 100 ns equilibration with 20 fs time steps, before 9 ×10 µs (complex membrane) or 5 ×10 µs (membrane without lipid I or lipid II) production simulations were run using 20 fs time steps, all in the NPT ensemble with the velocity-rescaling thermostat and semi-isotropic Parrinello–Rahman pressure coupling[34,35].

A pose of the *E. coli* MraY dimer with two lipid II molecules bound to the central cavity was selected for further analysis. All non-POPE lipids (except the two bound lipid II molecules) were deleted and the membrane allowed to shrink to 10 nm × 10 nm × 10.5 nm over 100 ns, with positional restraints applied to the protein backbone. The resulting molecule was then converted to the atomistic CHARMM36m force field[36,37] using the CG2AT2 protocol[38]. Side-chain pKas (negative log base 10 of the acid dissociation constant (Ka)) were assessed using propKa3.1 (ref. 39), and side-chain side charge states were set to their default. Production simulations were run for 5 repeats of ~510 ns, using a 2 fs time step in the NPT ensemble with the velocity-rescale thermostat and semi-isotropic Parrinello–Rahman pressure coupling[34,35].

All simulations were run in Gromacs (2021.3)[40]. Images were made in VMD[41]. Kinetic analysis of protein−lipid interactions and binding site identification were performed using PyLipID[42]. Density and contact analyses of atomistic MD simulations were performed using MDAnalysis[43,44]. Contacts were defined as a distance of less than 4 Å between lipid II and MraY.

### Expression and purification of the YES complex

The YES complex was expressed as described previously[27]. Briefly, Δ*slyD* BL21(DE3) competent cells were transformed with pET22b-SlyD$_{1–154}$ and pRSFDuet*Ec*MraY-E$_{ID21}$, and plated in LB agar containing 35 µg ml$^{-1}$ kanamycin and 100 µg ml$^{-1}$ ampicillin. The culture was grown in 2xYT media at 37 °C and 225 r.p.m., and induced at an OD$_{600}$ of 0.9 with 0.4 mM

IPTG at 18 °C overnight. The culture was collected by centrifugation at 9,000 × *g* for 10 min at 4 °C, followed by flash freezing.

The cells were lysed using an M-110L microfluidizer (Microfluidics) in 20 mM Tris-HCl pH 7.5, 300 mM NaCl, 10% glycerol, 5 mM *β*ME, 0.1 mM phenylmethyl sulfonyl fluoride and 0.1 mM benzamidine. The lysate was cleared by a 20-min centrifugation at 22,000 × *g*. The membrane was isolated by ultracentrifugation at 167,424 × *g* and solubilized in 10 mM HEPES pH 7.5, 300 mM NaCl, 5% glycerol, 5 mM *β*ME, 0.1 mM phenylmethyl sulfonyl fluoride, 0.1 mM benzamidine, 10 mM imidazole and 1% DDM. The extract was cleared by ultracentrifugation and then nutated with 1 ml NiNTA resin (Qiagen) at 4 °C for 2 h. The resin was washed with 5 CVs of wash buffer (10 mM HEPES pH 7.5, 150 mM NaCl, 5% glycerol, 5 mM *β*ME and 0.03% DDM) with 10 mM imidazole and eluted in 20 ml of wash buffer containing 200 mM imidazole. The eluent was further purified by size exclusion chromatography (Superdex 200 5/150 GL, Millipore Sigma) in 10 mM HEPES pH 7.5, 75 mM NaCl, 5% glycerol, 5 mM *β*ME and 0.03% DDM. Fractions were assessed by SDS–PAGE, concentrated and directly used for cryo-EM sample preparation.

### Sample preparation for cryo-EM
The protein sample was diluted to a concentration of 5 mg ml$^{-1}$ in 10 mM HEPES pH 7.5, 75 mM NaCl, 2% glycerol, 5 mM *β*ME, 0.03% DDM and 1 mM *E. coli* total lipid extract (Avanti Polar Lipids, 100600P). Quantifoil holey carbon films R1.2/1.3 300 mesh copper grids (Quantifoil, Micro Tools) were glow discharged with a 2 min 20 Å plasma current using a Pelco easiGlow, Emitech K100X. Grids were prepared using a Vitrobot system (FEI Vitrobot Mark v4 x2, Mark v.3) by applying 3 µl of 5 mg ml$^{-1}$ YES(T23P) complex onto the grid, followed by a 3.5 s blot using a +8-blot force and plunge frozen into liquid ethane.

### Data acquisition and analysis
Datasets were collected at ×105,000 magnification with a pixel size of 0.416 Å pixel$^{-1}$ using a 300 kV cryo-TEM Krios microscope equipped with a Gatan K3 6k × 4k direct electron detector and a Gatan energy filter (slit width 20 eV) in super-resolution mode using Serial EM. Movies with 40 frames were recorded with a total exposure dose of 60 e$^-$ Å$^{-2}$ and a defocus range of −1.0 to −2.5 µm. A total of 7,083 movies were gain reference and motion corrected using the patch motion correction built-in function in cryosparc (v.3.3.2) with a 2-fold bin that resulted in a pixel size of 0.832 Å pixel$^{-1}$ (ref. 45). The contrast transfer function (CTF) was estimated using CTFFIND4 (ref. 46). A total of 3,885,223 particles were obtained by template picker using PDB 8G01 as ref. 27. Four ab-initio models were obtained using 500,000 particles, from which the best and worst volumes were used to sort 4x binned particles through heterogeneous refinement.

Iterative rounds of heterogeneous and non-uniform refinement were performed before re-extracting particles using a 2x bin. This process was continued and the resulting particles were re-extracted using a 1.3x bin. After several rounds of heterogeneous and non-uniform refinement, 575,243 particles were extracted without binning and used to create a map through non-uniform refinement. Using the MraY model from PDB 8G01, a mask covering only the density encompassing MraY was created using ChimeraX[47]. Density outside of this mask was removed using particle subtraction, followed by ab-initio modelling. The best fitting map was then used for further refinement using global CTF, heterogeneous and non-uniform refinement. The final map with a 3.8 Å resolution was composed by 287,765 particles and sharpened using the autosharpen module in PHENIX-1.19.2. The data collection, refinement and validation statistics can be found in Supplementary Table 2.

### Statistics and reproducibility
No statistical methods were used to predetermine sample sizes used for experiments, but sample sizes are in line with field standards. No data were excluded from the analyses. The experiments were not randomized. The investigators were not blinded to allocation during experiments and outcome assessment.

### Reporting summary
Further information on research design is available in the Nature Portfolio Reporting Summary linked to this article.

## Data availability
All bacterial strains and plasmids developed in this study are available upon request. The atomic coordinates presented in this study have been deposited in the RSCB Protein Data Bank under the accession number PDB 8TLU. Source data are provided with this paper.

## Code availability
No code was used to analyse data in this study.

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

## Acknowledgements

We thank all members of the Bernhardt, Clemons, Stansfeld and Rudner Labs for thoughtful discussions and advice throughout this project; the members of the CMCB (Center for Microbial Chemical Biology) team at McMaster University, and the Andres and Whitney labs at McMaster University for use of their lab space and facilities by L.S.M.; and L. Burrows for the gift of the B-band serotype O5 monoclonal antibody. This work was supported by the National Institutes of Health (AIO83365 to T.G.B. and R01GM114611 to W.M.C.), The G. Harold and Leila Y. Mathers Foundation (to W.M.C.), Wellcome Trust (208361/Z/17/Z to P.J.S.), MRC (MR/S009213/1 to P.J.S.), BBSRC (BB/P01948X/1, BB/R002517/1 and BB/S003339/1 to P.J.S.), the Howard Dalton Centre (to P.J.S.), and Investigator funds from the Howard Hughes Medical Institute (to T.G.B.). L.S.M. was supported by an NSERC Postdoctoral fellowship. D.S. was supported by a CIHR postdoctoral fellowship. A.F.-G. was supported by an NSERC USRA. B.W.A.B.'s studentship was sponsored by the MRC. (Cryo)electron microscopy by W.M.C.'s lab was done in the Beckman Institute Resource Center for Transmission Electron Microscopy at Caltech with help from S. Chen. MD analysis in P.J.S.'s group made use of time on ARCHER and JADE granted via the UK High-End Computing Consortium for Biomolecular Simulation, HECBioSim (https://www.hecbiosim.ac.uk/), supported by EPSRC (grant no. EP/R029407/1). P.J.S. acknowledges Sulis at HPC Midlands+, which was funded by the EPSRC on grant EP/T022108/1, and the University of Warwick Scientific Computing Research Technology Platform for computational access.

## Author contributions

L.S.M. performed molecular cloning, purification of *Pa*MraY, in vitro assays, assessment of mutant phenotypes and isolation of

lipid II. D.S. assisted with LC–MS methods and assay development. A.F.-G. assisted with site-directed mutagenesis and spot dilution assays. A.K.O., Y.E.L. and W.M.C. determined the cryo-EM structure of $^{Ec}$MraY and $^{Ec}$MraY(T23P). R.A.C., B.W.A.B. and P.J.S. designed, performed and analysed molecular dynamics simulations. N.G.G. conducted the original suppressor screen and generated *P. aeruginosa* strains. Overall project supervision was performed by T.G.B. The manuscript was written by L.S.M. and T.G.B. All other authors edited and approved the manuscript.

## Competing interests

The authors declare no competing interests.

## Additional information

**Extended data** is available for this paper at https://doi.org/10.1038/s41564-024-01603-2.

**Correspondence and requests for materials** should be addressed to Thomas G. Bernhardt.

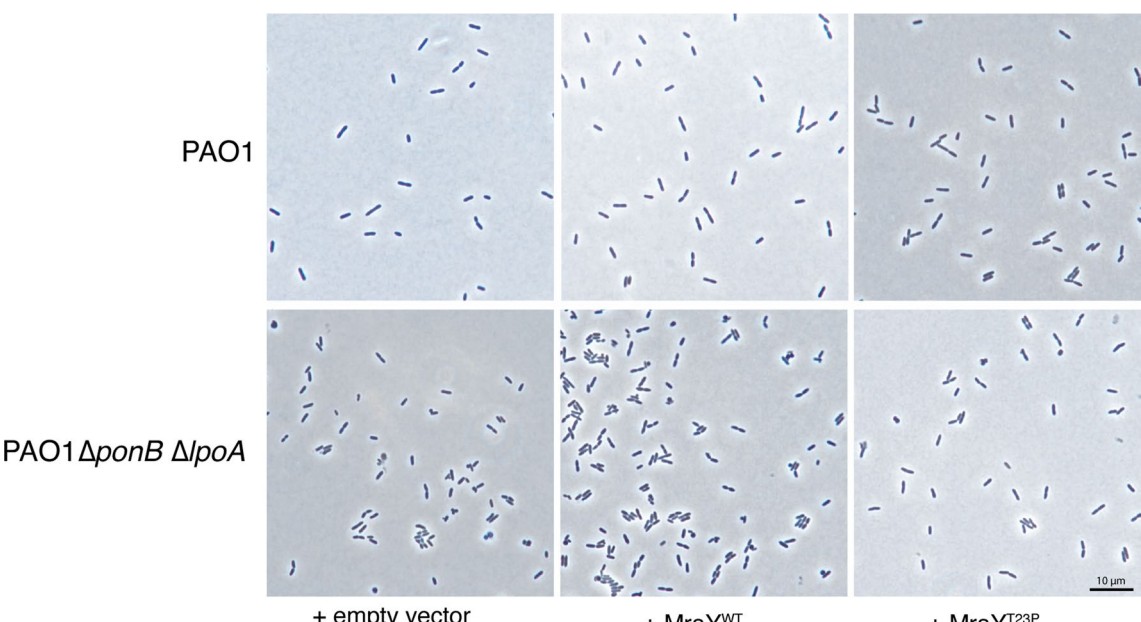

**Extended Data Fig. 1 | Expression of *Pa*MraY(T23P) in *P. aeruginosa* Δ*ponB* Δ*lpoA* rescues cell shape defects.** Phase contrast micrographs of *P. aeruginosa* PAO1 and PAO1 Δ*ponB* Δ*lpoA* cells grown in LB with 1 mM IPTG to induce the indicated MraY protein. Scale bar = 10 μm. Representative images of two independent experiments are shown.

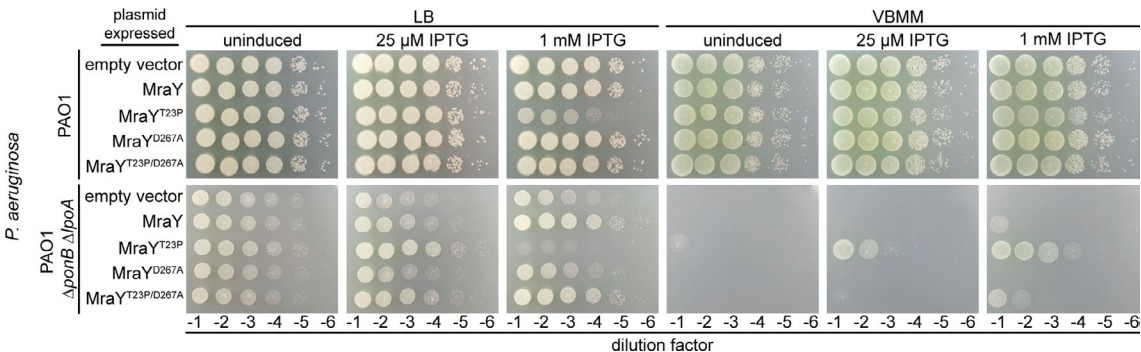

**Extended Data Fig. 2 | Catalytic activity is required for MraY(T23P) to suppress cell wall defects.** Ten-fold serial dilutions of cells of the indicated *P. aeruginosa* strains harboring expression plasmids producing the indicated MraY variant were plated on media with or without IPTG to induce production of MraY variants as indicated.

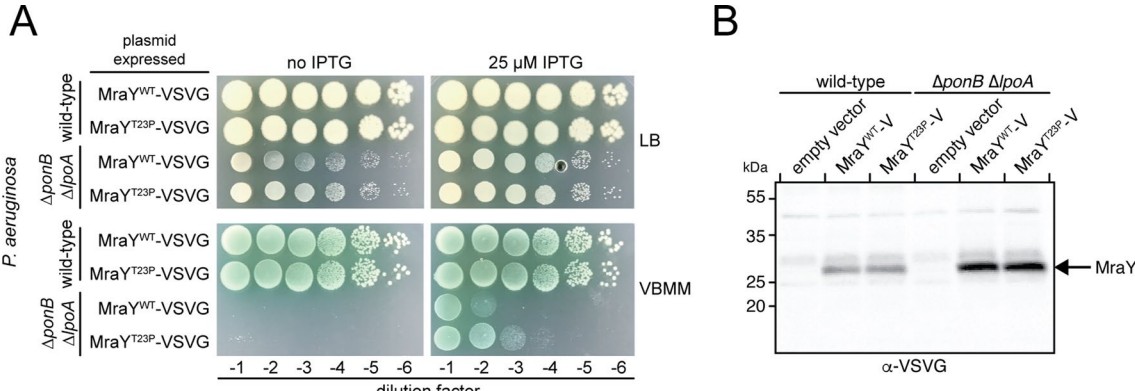

**Extended Data Fig. 3 | Cells produce MraY(WT) and MraY(T23P) to comparable levels.** (**a**) Ten-fold serial dilutions of *P. aeruginosa* cells harboring expression plasmids producing the indicated VSVG-tagged MraY were plated on media with or without inducer as indicated. (**b**) Western blot of cells expressing MraY(WT)-VSVG or MraY(T23P)-VSVG. *P. aeruginosa* cells expressing the indicated plasmid were grown to mid-log, normalized for optical density, and extracts were prepared for immunoblotting. Protein was detected using α-VSVG antibody. Data is representative of two replicates.

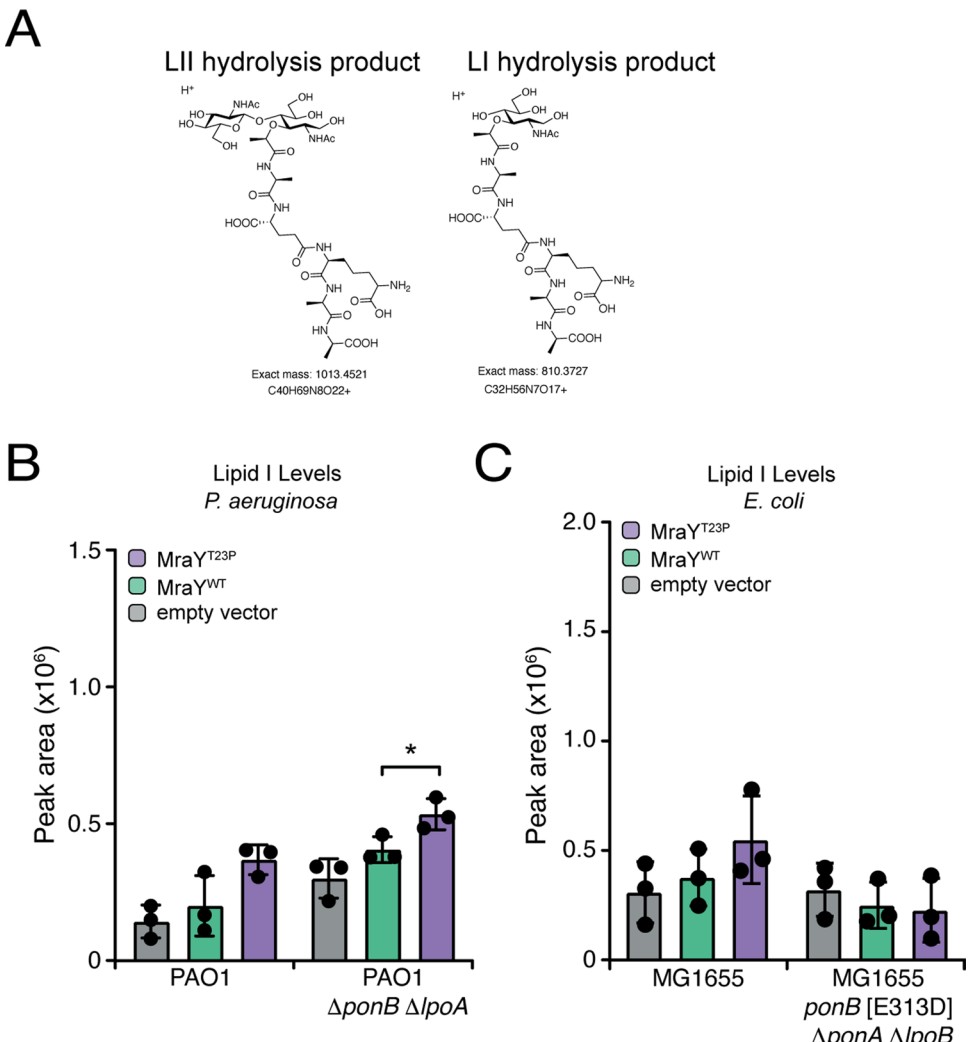

**Extended Data Fig. 4 | Lipid I levels in cells producing MraY(WT) or MraY(T23P).** (a) Chemical structures of the Lipid II (LII) and Lipid I (LI) hydrolysis products detected by LCMS. Quantification of extracted ion chromatograms of the lipid I hydrolysis product for the indicated *P. aeruginosa* (**b**) and *E. coli* (**c**) strains. Three independent extractions were performed with lipid I levels quantified using the area of the peak from the extracted ion chromatogram using the Agilent software. Dots represent the values obtained for the biological replicates and the bars indicate the mean. Error bars represent SD. For MraY(T23P) vs MraY(WT) in PAO1 Δ*ponB* Δ*lpoA* *$P$ = 0.039, in PAO1, MG1655, MG1655 Δ*ponA* Δ*lpoB ponB*[E313D], not significant (unpaired, two-tailed, t-test).

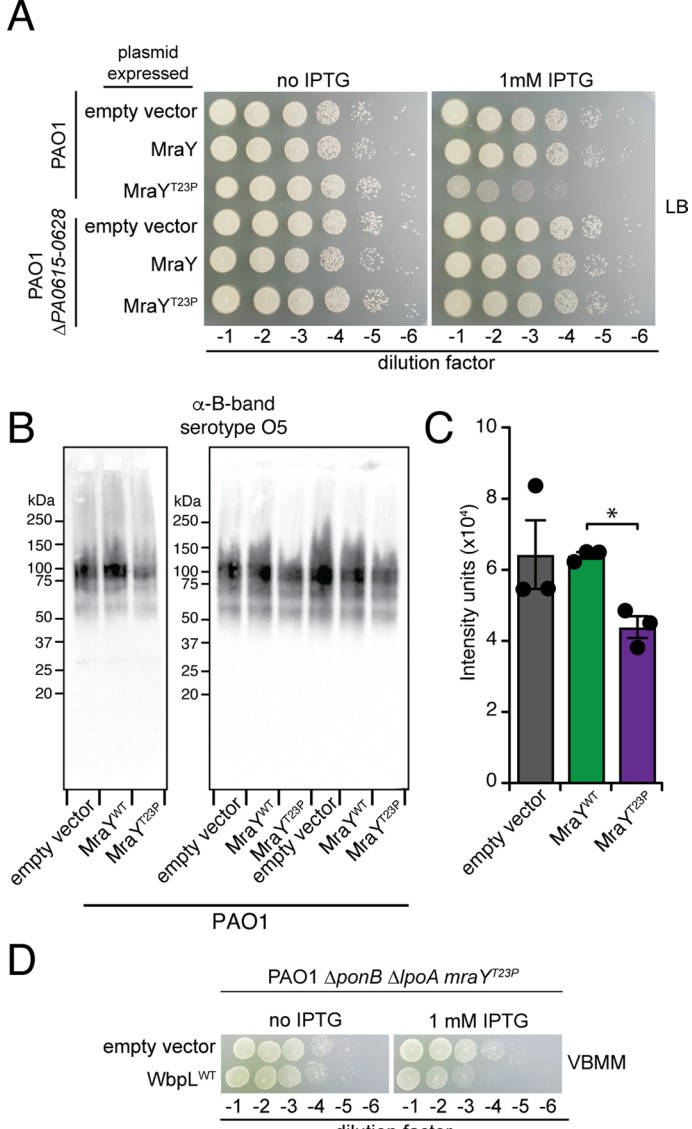

**Extended Data Fig. 5 | Expression of *Pa*MraY(T23P) causes a pyocin-dependent growth defect in *P. aeruginosa* due to a reduction in O-antigen production.** (**a**) Ten-fold serial dilutions of *P. aeruginosa* strains harboring expression plasmids producing the indicated MraY variant were plated on LB containing with or without IPTG to induce protein production from the plasmids. (**b**) Western blots of B-band O-antigen from *P. aeruginosa* cells expressing the MraY proteins as indicated. Image contains three independent experiments. (C) The B-band LPS from three independent replicates of sample extraction was quantified using densitometry. Dots represent the values obtained for the biological replicates and the bars indicate the mean. Error bars represent SEM, *P = 0.0031 (unpaired, two-tailed, t-test). (**d**) Ten-fold serial dilutions of *P. aeruginosa* cells harboring expression plasmids producing the indicated WbpL protein, dilutions were plated on VBMM with or without IPTG to induce the WbpL protein as indicated. Abbreviations: WT, wild-type; VBMM, Vogel⁻Bonner minimal medium; IPTG, isopropyl-B-D-1-thiogalactopyranoside.

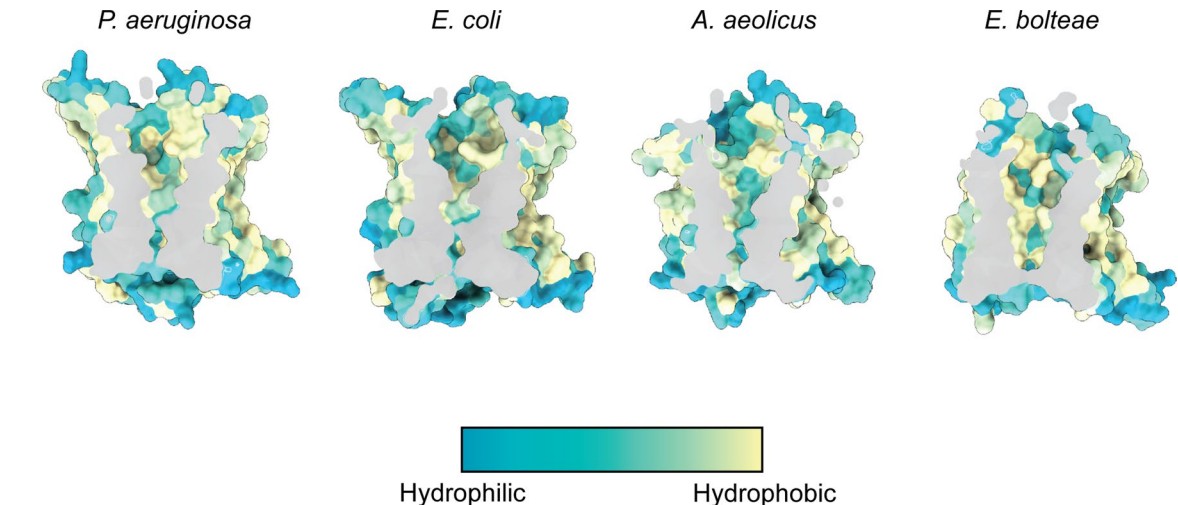

**Extended Data Fig. 6 | The cavity of MraY is hydrophobic.** Hydrophobic surface representation of the structures of MraY from *E. coli* (PDB 8G01), *A. aeolicus* (PDB 4J72), *E. boltae* (PDB 5JNQ), and the Alphafold2 model of *P. aeruginosa* MraY, colored according to the scale as shown in the Figure.

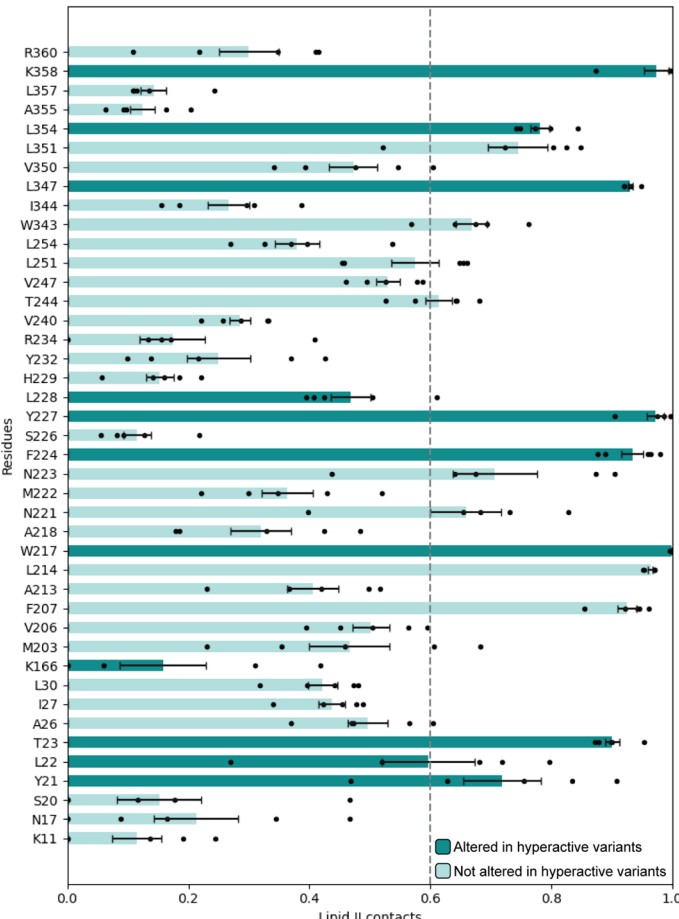

**Extended Data Fig. 7 | MraY residues contacting lipid II in the MD simulations.** Lipid II contacts with MraY residues from atomistic MD simulations. Error bars represent standard error from 5 repeats. Darker green bars represent residues altered in hyperactive variants. Dashed line at x = 0.6 represents cutoff for interactions shown in Fig. 4c.

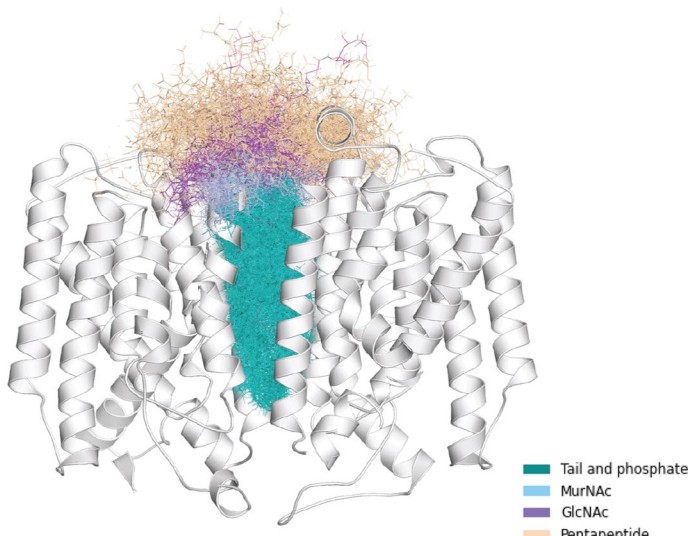

Tail and phosphate
MurNAc
GlcNAc
Pentapeptide

**Extended Data Fig. 8 | Flexibility of MraY bound lipid II in the MD simulation.** All states of lipid II from 5 repeats of atomistic simulation overlaid onto the structure of MraY. Colored as in Fig. 4d.

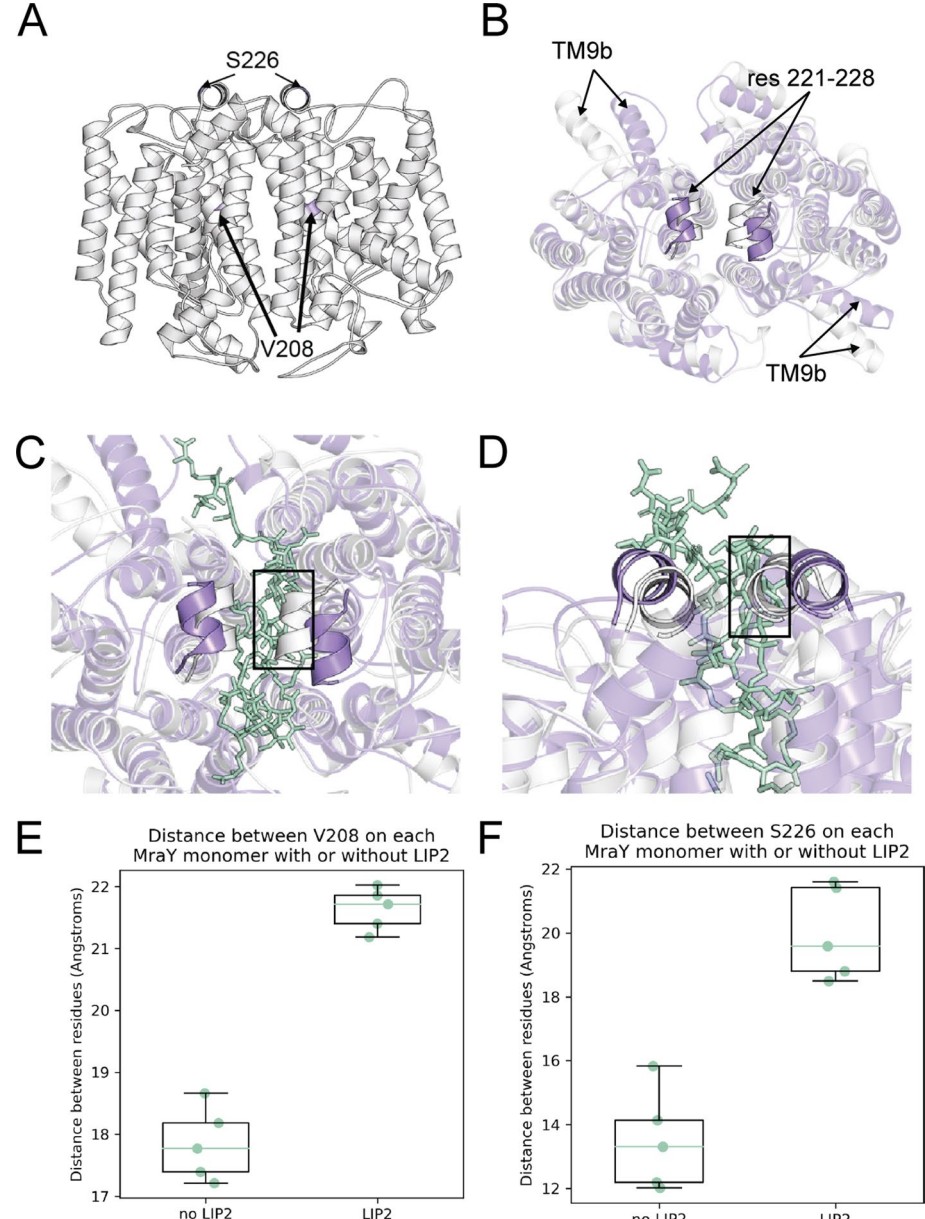

**Extended Data Fig. 9 | MD analysis identifies potential conformational changes in MraY upon lipid II binding. (a)** Structure of MraY dimer in state when lipid II is bound (not shown). Residues V208 and S226 are indicated and colored purple. **(b-d)** An overlay of the structure of MraY at the end of simulations with (purple) or without (gray) lipid II present. **(b)** The structure is shown from the top, lipid II is hidden, and helices with notable differences are indicated. **(c, d)** MraY with lipid II, boxes indicate where lipid II clashes with the structure from the simulation without lipid II, indicating why the periplasmic helix 221-228 is moved apart when lipid II is bound. **(c)** is top (periplasmic) view, while **(d)** is a side view. **(e)** A boxplot of the average distance between V208 (a residue in the lipid II binding pocket) of each MraY monomer, in simulations

with or without lipid II present. The data represented by each box plot is the mean distance from all time points in each of 5 repeats (minima/maxima: 17.2/18.7, no lip2; 21.2/22.0, lip2). Box plot center line represents the median, while the box limits represent the upper and lower quartiles. The whiskers represent the 1.5x interquartile range. **(f)** A boxplot of the average distance between S226 (a residue in the periplasmic helix above the lipid II binding site) of each MraY monomer, in simulations with or without lipid II present. The data represented by each box plot is the mean distance from all time points in each of 5 repeats (minima/maxima: 12.0/15.8, no lip2; 18.5/21.6, lip2). Box plot center line represents the median, while the box limits represent the upper and lower quartiles. The whiskers represent the 1.5x interquartile range.

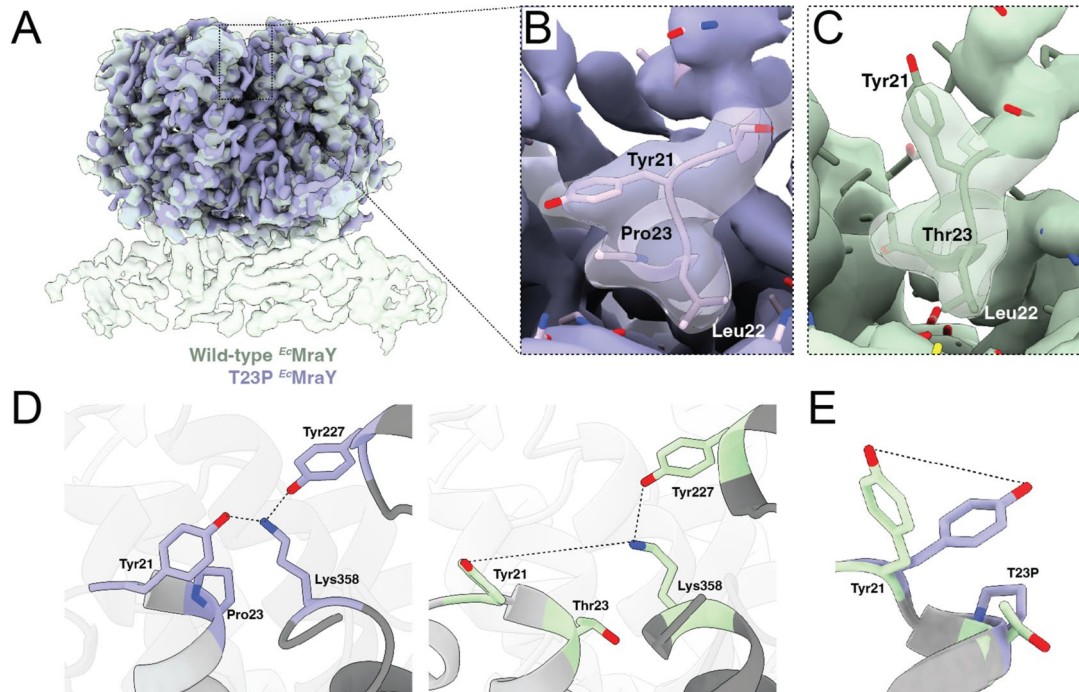

**Extended Data Fig. 10 | Comparison of MraY(WT) and MraY(T23P) structures in the YES complex.** (**a**) Overlay of densities of MraY(WT) (EMD-29641) (green) and MraY(T23P) (purple) viewed in the plane of the membrane. (**b**) Enlarged view of the densities around the T23P mutant. Residues are shown in stick representation. Residues 21-23 are labeled for reference. (**c**) As in *B* for the wild-type complex. (**d**) Hydrogen bonding network observed in MraY(T23P) (left, purple) compared to WT (right, green) at the mutagenesis site involving Y21, Y227, and K358. (**e**) Similar to *D*, overlay of the two models highlighting the conformational differences of residue Y21.

# Reporting Summary

## Statistics

For all statistical analyses, confirm that the following items are present in the figure legend, table legend, main text, or Methods section.

| n/a | Confirmed | |
|---|---|---|
| ☐ | ☒ | The exact sample size (*n*) for each experimental group/condition, given as a discrete number and unit of measurement |
| ☐ | ☒ | A statement on whether measurements were taken from distinct samples or whether the same sample was measured repeatedly |
| ☐ | ☒ | The statistical test(s) used AND whether they are one- or two-sided *Only common tests should be described solely by name; describe more complex techniques in the Methods section.* |
| ☒ | ☐ | A description of all covariates tested |
| ☒ | ☐ | A description of any assumptions or corrections, such as tests of normality and adjustment for multiple comparisons |
| ☐ | ☒ | A full description of the statistical parameters including central tendency (e.g. means) or other basic estimates (e.g. regression coefficient) AND variation (e.g. standard deviation) or associated estimates of uncertainty (e.g. confidence intervals) |
| ☐ | ☒ | For null hypothesis testing, the test statistic (e.g. *F*, *t*, *r*) with confidence intervals, effect sizes, degrees of freedom and *P* value noted *Give P values as exact values whenever suitable.* |
| ☒ | ☐ | For Bayesian analysis, information on the choice of priors and Markov chain Monte Carlo settings |
| ☒ | ☐ | For hierarchical and complex designs, identification of the appropriate level for tests and full reporting of outcomes |
| ☒ | ☐ | Estimates of effect sizes (e.g. Cohen's *d*, Pearson's *r*), indicating how they were calculated |

*Our web collection on statistics for biologists contains articles on many of the points above.*

## Software and code

Policy information about availability of computer code

| Data collection | SerialEM Version 4.1 (CryoEM) |
|---|---|
| Data analysis | Statistical analyses were performed by Prism 9 (GraphPad Software, LLC.). MD simulations were run in Gromacs 2021.3. Images were made in VMD. Kinetic analysis of protein-lipid interactons and binding site identification were performed using PyLipID. Density and contact analyses of atomistic MD simulations were performed using MDAnalysis. LC/MS data were analyzed using Agilent MassHunter Workstation Qualitative Analysis software. Structural data were visualized using ChimeraX 1.6.1. The final map was generated using PHENIX-1.19.2. Data processing was performed using cryosparc V3.3.2. The contrast transfer function (CTF) was estimated using CTFFIND4 4.1.10 . |

For manuscripts utilizing custom algorithms or software that are central to the research but not yet described in published literature, software must be made available to editors and reviewers. We strongly encourage code deposition in a community repository (e.g. GitHub). See the Nature Portfolio guidelines for submitting code & software for further information.

## Data

Policy information about availability of data

All manuscripts must include a data availability statement. This statement should provide the following information, where applicable:

- Accession codes, unique identifiers, or web links for publicly available datasets
- A description of any restrictions on data availability
- For clinical datasets or third party data, please ensure that the statement adheres to our policy

All bacterial strains and plasmids developed in this study are available upon request. The atomic coordinates presented in this study have been deposited in the RSCB Protein Data Bank (PDB) under the accession number PDB: 8TLU. Previously published structures referenced in this manuscript can be accessed in the PDB using the accession numbers: 8G01 (E. coli MraY), 4J72 (A. aeolicus MraY), 5JNQ (E. boltae MraY).

## Research involving human participants, their data, or biological material

Policy information about studies with human participants or human data. See also policy information about sex, gender (identity/presentation), and sexual orientation and race, ethnicity and racism.

| | |
|---|---|
| Reporting on sex and gender | N/A |
| Reporting on race, ethnicity, or other socially relevant groupings | N/A |
| Population characteristics | N/A |
| Recruitment | N/A |
| Ethics oversight | N/A |

Note that full information on the approval of the study protocol must also be provided in the manuscript.

# Field-specific reporting

Please select the one below that is the best fit for your research. If you are not sure, read the appropriate sections before making your selection.

☒ Life sciences   ☐ Behavioural & social sciences   ☐ Ecological, evolutionary & environmental sciences

For a reference copy of the document with all sections, see nature.com/documents/nr-reporting-summary-flat.pdf

# Life sciences study design

All studies must disclose on these points even when the disclosure is negative.

| | |
|---|---|
| Sample size | No statistical methods were used to predetermine sample size for experiments, but sample sizes are in line with field standards. All experiments were performed at least twice. |
| Data exclusions | No data were excluded from this study. |
| Replication | All data are from a minimum of two independent experiments. All replication attempts were successful. |
| Randomization | Samples were not randomized in this study. Covariates were controlled by reproducing the experiments on separate days. |
| Blinding | We did not blind samples as, after each experimental setup, all measurements and analyses were performed identically across all conditions. |

# Reporting for specific materials, systems and methods

We require information from authors about some types of materials, experimental systems and methods used in many studies. Here, indicate whether each material, system or method listed is relevant to your study. If you are not sure if a list item applies to your research, read the appropriate section before selecting a response.

## Materials & experimental systems

| n/a | Involved in the study |
|-----|----------------------|
| ☐ | ☒ Antibodies |
| ☒ | ☐ Eukaryotic cell lines |
| ☒ | ☐ Palaeontology and archaeology |
| ☒ | ☐ Animals and other organisms |
| ☒ | ☐ Clinical data |
| ☒ | ☐ Dual use research of concern |
| ☒ | ☐ Plants |

## Methods

| n/a | Involved in the study |
|-----|----------------------|
| ☒ | ☐ ChIP-seq |
| ☒ | ☐ Flow cytometry |
| ☒ | ☐ MRI-based neuroimaging |

# Antibodies

| | |
|---|---|
| Antibodies used | Anti-VSVG (Sigma, V4888); anti-Rabbit IgG HRP (7074S, NEB); P. aeruginosa serotype O5 mouse monoclonal antibody (MediMabs, MM-76605-1; Clone ID MF15-4); anti-Mouse HRP (NEB, 7076S) |
| Validation | The Anti-VSVG antibody has been used numerous times to detect VSVG-tagged proteins by western blot and validation information can be found at the manufacturers' website (https://www.sigmaaldrich.com/US/en/product/sigma/v4888). In addition, we verified that the antibody does not react with Pseudomonas aeruginosa lysates derived from strains that do not encode a VSVG-tagged protein (see figure S2). P. aeruginosa serotype O5 mouse monoclonal antibody has been validated previously to detect O-antigen from P. aeruginosa serotype O5 (https://link.springer.com/protocol/10.1007/978-1-4939-0473-0_31). |

