## [Peer Review File · Nature Microbiology]

Peer Review Information

Journal: Nature Microbiology

Manuscript Title: Synthesis of lipid-linked precursors of the bacterial cell wall is governed by a feedback control mechanism in *Pseudomonas aeruginosa*

Corresponding author name(s): Dr Thomas Bernhardt

Reviewer Comments & Decisions:

Decision Letter, initial version:

Message: 19th September 2023

Dear Dr Bernhardt,

Thank you for your patience while your manuscript "A feedback control mechanism governs the synthesis of lipid-linked precursors of the bacterial cell wall" was under peer-review at Nature Microbiology. It has now been seen by 3 referees, whose expertise and comments you will find at the end of this email. Although they find your work of some potential interest and are very enthusiastic about your work, they have nevertheless raised a number of points that will need to be addressed before we can consider publication of the work in Nature Microbiology. Overall, however, the points raised by the referees are very clear, and they should, we hope, be straightforward to address with some simple experiments.

Should further experimental data allow you to address these criticisms, we would be happy to look at a revised manuscript.

Please include a data availability statement as a separate section after Methods but before references, under the heading "Data Availability". This section should inform readers about the availability of the data used to support the conclusions of your study. This information includes accession codes to public repositories (data banks for protein, DNA or RNA sequences, microarray, proteomics data etc...), references to source data published alongside the paper, unique identifiers such as URLs to data repository entries, or data set

2DOIs, and any other statement about data availability. At a minimum, you should include the following statement: "The data that support the findings of this study are available from the corresponding author upon request", mentioning any restrictions on availability. If DOIs are provided, we also strongly encourage including these in the Reference list (authors, title, publisher (repository name), identifier, year). For more guidance on how to write this section please see: <http://www.nature.com/authors/policies/data/data-availability-statements-data-citations.pdf>

* If you have not done so already we suggest that you begin to revise your manuscript so that it conforms to our Article format instructions at <http://www.nature.com/nmicrobiol/info/final-submission>. Refer also to any guidelines provided in this letter.

When submitting the revised version of your manuscript, please pay close attention to our [href="https://www.nature.com/nature-portfolio/editorial-policies/image-integrity">Digital Image Integrity Guidelines. and to the following points below:](https://www.nature.com/nature-portfolio/editorial-policies/image-integrity)

Note: This url links to your confidential homepage and associated information about manuscripts you may have submitted or be reviewing for us. If you wish

2to forward this e-mail to co-authors, please delete this link to your homepage first.

Nature Microbiology is committed to improving transparency in authorship. As part of our efforts in this direction, we are now requesting that all authors identified as 'corresponding author' on published papers create and link their Open Researcher and Contributor Identifier (ORCID) with their account on the Manuscript Tracking System (MTS), prior to acceptance. This applies to primary research papers only. ORCID helps the scientific community achieve unambiguous attribution of all scholarly contributions. You can create and link your ORCID from the home page of the MTS by clicking on 'Modify my Springer Nature account'. For more information please visit please visit www.springernature.com/orcid.

If you wish to submit a suitably revised manuscript we would hope to receive it within 6 months. If you cannot send it within this time, please let us know. We will be happy to consider your revision, even if a similar study has been accepted for publication at Nature Microbiology or published elsewhere (up to a maximum of 6 months).

Yours sincerely,

Reviewer Expertise:

Referee #1: membrane proteins, structural biology, cryoEM

Referee #2: cell wall biosynthesis, lipid metabolism

Referee #3: cell wall biosynthesis, lipid metabolism

Reviewer Comments:

Reviewer #1 (Remarks to the Author):

Summary

Marmont et al. present an in-depth investigation into a previously identified mutation that was able to suppress a conditionally lethal defect in aPBP activity in *Pseudomonas aeruginosa*. Using an aPBP deficient strain, the authors show that a T23P mutation in *MraY* can rescue a growth defect on minimal media, and clearly demonstrate that the rescue is due to altered *MraY* activity rather than increased accumulation of a faulty enzyme. This result is replicated in *Escherichia coli* using a similar aPBP deficient strain, indicating there is a conserved mechanism at play. The authors go on to show that the overproduction of *MraY*(T23P) leads to an increase in lipid II levels in cells, and that the purified mutant enzyme is significantly more active in a biochemical assay than the wildtype form. After showing that overproduction of *MraY*(T23P) disrupts proper O-antigen synthesis, leading to R2-pyocin self-killing, the authors go on to propose a structural rationale for the mutant phenotype that centered on a hydrophobic cavity at the dimer interface of *MraY*. A previous

3cryo-EM structure with unexplained density in the cavity suggested it could accommodate a molecule of lipid II, and focused refinement of the *E. coli* *MraY*(T23P) component of the YES complex cryo-EM structure was consistent with that hypothesis. Finally, the authors used molecular dynamics simulations to assess whether a lipid II molecule could enter the cavity, using the *E. coli* *MraY* structure from the previously published YES complex structure. The simulations supported the authors' hypothesis and allowed them to propose an elegant feedback mechanism whereby accumulation of lipid II acts to downregulate the peptidoglycan biosynthesis pathway, allowing for limited carrier lipid to be distributed among the various biosynthetic pathways that rely upon it.

Overall, this is a fantastic piece of research and leads one to wonder about the existence of similar feedback mechanisms in teichoic acid, O-antigen, and other cell envelope polymers. The identification of the feedback mechanism is clearly the outstanding result and I think it will have an immediate impact on the field of bacterial cell wall research.

Minor Points

1. Though the authors do acknowledge that it is unclear how the T23P mutation leads to increased *MraY* activity, I wonder if running the MD simulations using the mutant of interest might have provided some insight?
2. Densitometry measurements for quantitation are not ideal, and it is difficult for me to clearly see that the *MraY*(T23P) O-antigen blot is significantly less intense than the wildtype or vector. It would be nice to see images of the three full gels/blots in the supplementary material.
3. "Using course-grained MD simulations" on Line 291 should be changed to "coarse".

Reviewer #2 (Remarks to the Author):

The bacterial cell envelope is an essential structure that resists antibiotics, immune systems, and environmental hazards. Bacterial glycoconjugates are foundational for envelope assembly and include the peptidoglycan (PG) cell wall, LPS O-antigens, capsules, and other clinically important structures. The glycan moiety of nearly all bacterial glycoconjugates is assembled on a lipid carrier known as undecaprenyl phosphate (Und-P). Since most bacteria encode multiple Und-P-utilizing pathways, the pool of Und-P must be shared between pathways. But how? While the mechanisms that regulate Und-P distribution are not known, the present work by Marmont et al. indicates that one mechanism involves feedback inhibition. Briefly, the authors isolated suppressors encoding an altered *MraY* enzyme (T23P mutation) that restores growth to *P. aeruginosa* cells defective for PG biosynthesis. Subsequent work confirmed that *MraY*(T23P) restores growth to *P. aeruginosa*, as well as *E. coli*, cells defective for PG biosynthesis. A closer examination revealed that cells expressing *MraY*(T23P) accumulate PG intermediates (lipid II), indicating that *MraY*(T23P) is a hyperactive enzyme. Since *MraY* competes with other initiating glycosyltransferases for Und-P, the authors wondered if misregulated *MraY* activity would impede the production of other Und-P-dependent polymers. Indeed, O-antigen production was reduced in cells expressing *mraY*(T23P). This finding suggests that the kinetics of

4

initiating glycosyltransferases like *MraY* (PG) or *WbpL* (O-antigen) play a major role in regulating Und-P distribution. Finally, structural studies indicated the extracytoplasmic side of *MraY* is the regulatory site for this enzyme and that lipid II binding likely induces conformational changes in *MraY* that affect its activity.

Overall, the manuscript is well-written, the experiments well-controlled, and the conclusions appropriate. The finding that O-antigen expression is reduced in cells expressing *mraY*(T23P) is especially intriguing. It would be interesting to know if hyperactive variants of *WbpL* reduce PG production in *P. aeruginosa* in a similar manner. More generally, the discovery that feedback inhibition alters competition for Und-P has important implications for understanding how cell envelope assembly is coordinated. I have only a few minor comments.

Reversible reactions

Lines 432-437: Can the feedback mechanism help explain why initiating glycosyltransferases like *MraY* catalyze reversible reactions (e.g., PMID: 14324547, 3034883, 318640, 16237026)?

On a related note, can the authors comment on whether they think their hyperactive variants of *MraY* are capable of catalyzing the reverse formation of lipid I?

Multicopy suppression

In Figure 1, the authors show that *MraY*(T23P) suppresses the growth defect of cells lacking *PonB* and *LpoA*. Since *PonB* and *LpoA* are required for PG biosynthesis, it would be helpful to see the shape of these cells.

Figure 2 and S3

Is there a reason significance was not plotted in Figs. 2 and S3? It is a little confusing to understand significance as described in the legends of these figures.

Reviewer #3 (Remarks to the Author):

The manuscript by Marmont and colleagues entitled "A feedback control mechanism governs the synthesis of lipid-linked precursors of the bacterial cell wall" provides new information on how bacterial cells balance the utilization of the precious resource C55P/PP. This lipid-based carrier molecule can be decorated with glycans for the subsequent use in vastly distinct components of the cell envelope. This includes periplasmic peptidoglycan, as well extracellularly polysaccharides in the form of O-antigen. Although there are likely to be multiple control points across various stages of these biochemical pathways, the authors have specifically interrogated when C55 is linked to a GlcNAc-MurNAc-pentapeptide (lipid II), the final precursors for peptidoglycan biosynthesis. The team of researchers capitalised on their previous findings on this pathway. In this study, they designed a series of assays to illustrate that that mutation of T23 in *MraY* leads to the protein being resistant to Lipid II accumulation, and the subsequent starvation of C55 in other cellular aspects, eg *Oag*. Using a combination of cryoEM and in silico analyses, the authors show that Lipid II binding in *MraY* occurs at the periplasmic face of the IM, and that the T23P mutation is likely to

5

obliterate this binding capacity. Since the structural analyses indicate that binding of flipped (periplasmic facing) Lipid II can lead to allosteric modification in the protein's active site, the T23P mutant retains activity, despite Lipid II hyper-accumulation.

Comments:

The phenotypic analyses are somewhat indirect. Additional whole cell assays to back up the findings in *Ec* and *Pa* are desired. Direct co-purification of Lipid II-WT *MraY* vs Lipid II-*MraYT23P* would provide stronger evidence.

Alternative means of illustrating the significance of the *MraY* periplasmic lipid II binding requirements, other than mutation of *ponA/B_lpoA/B* are desired. For example, specific conditions (eg antimicrobials) that alter PG vs Oag demands will place a greater emphasis on the biological dynamics of this feedback loop.

A competitive MD analysis of its cytoplasmic enzymatic role, and Lipid II periplasmic sequestering may shed light on the proposed direct inhibition model.

Line 425-437 are weak, thereby impacting the relevance of Line 87-97. Expansion of knowledge by including bioinformatics on more species/across biology is appropriate if the authors want to retain the broader significant claims in the introduction and discussion.

Line 188: Normalization by optical density is not adequate as the sole means to define inputs, as these values can be influenced by many factors which would skew the data (eg cell size/morphology, capsule production). Please combine with eg CFU, microscopy, protein quant.

Line 358: It levels off in the WT, that is a measure of activity and then loss of activity, hence the inability to inhibit WT *MraY* activity needs to be addressed.

Line 381- 406: These sections appear more appropriate for the results, not discussion.

Figure 3: Please discuss the role of K166, as it is not directly involved in binding, could it be Lipid II recruitment?

Are the hydrophobic residues lining the cavity highly conserved across *Pseudomonas* spp, or in other genera?

Lines 90 & 91: Change "produce" to for example "carry"

Line 193: Why are Lipid II levels not higher in the *E. coli* mutant with *MraYT23P* overexpression? This section is very succinct, and details are missing.

Line 228: "Strikingly" is an overstatement.

Line 305: Only W217 is at 100%

Line 325: Reword "the first membrane step"

Figure 2: Embed stats in bar graphs.

Figure 4: Improve quality of the x- and y-labels.

Author Rebuttal to Initial comments

Reponses to Reviewer Comments

Reviewer #1 (Remarks to the Author):

Summary

Marmont et al. present an in-depth investigation into a previously identified mutation that was able to suppress a conditionally lethal defect in aPBP activity in *Pseudomonas aeruginosa*. Using an aPBP deficient strain, the authors show that a T23P mutation in *MraY* can rescue a growth defect on minimal media, and clearly demonstrate that the rescue is due to altered *MraY* activity rather than increased accumulation of a faulty enzyme. This result is replicated in *Escherichia coli* using a similar aPBP deficient strain, indicating there is a conserved mechanism at play. The authors go on to show that the overproduction of *MraY*(T23P) leads to an increase in lipid II levels in cells, and that the purified mutant enzyme is significantly more active in a biochemical assay than the wildtype form. After showing that overproduction of *MraY*(T23P) disrupts proper O-antigen synthesis, leading to R2-pyocin self-killing, the authors go on to propose a structural rationale for the mutant phenotype that centered on a hydrophobic cavity at the dimer interface of *MraY*. A previous cryo-EM structure with unexplained density in the cavity suggested it could accommodate a molecule of lipid II, and focused refinement of the *E. coli* *MraY*(T23P) component of the YES complex cryo-EM structure was consistent with that hypothesis. Finally, the authors used molecular dynamics simulations to assess whether a lipid II molecule could enter the cavity, using the *E. coli* *MraY* structure from the previously published YES complex structure. The simulations supported the authors' hypothesis and allowed them to propose an elegant feedback mechanism whereby accumulation of lipid II acts to downregulate the peptidoglycan biosynthesis pathway, allowing for limited carrier lipid to be distributed among the various biosynthetic pathways that rely upon it.

Overall, this is a fantastic piece of research and leads one to wonder about the existence of similar feedback mechanisms in teichoic acid, O-antigen, and other cell envelope polymers. The identification of the feedback mechanism is clearly the outstanding result and I think it will have an immediate impact on the field of bacterial cell wall research.

7RESPONSE: We thank the reviewer for their enthusiasm for our work.

Minor Points

1. Though the authors do acknowledge that it is unclear how the T23P mutation leads to increased MraY activity, I wonder if running the MD simulations using the mutant of interest might have provided some insight?

RESPONSE: The reviewer makes a great suggestion. We indeed ran a coarse-grained MD simulation using *Escherichia coli* MraY^{T23P}. Unfortunately, we did not observe any detectable difference in lipid binding or overall protein conformation compared to the wild-type protein that might provide insights into the regulatory mechanism. However, coarse-grained simulations typically do not model the effects of single-site substitutions on overall protein conformation very well. Also, based on the short time frame of the simulations, even a 10-fold reduction would likely look fairly similar in the analysis. Given the inconclusiveness of these simulations, we did not include them in the manuscript.

2. Densitometry measurements for quantitation are not ideal, and it is difficult for me to clearly see that the MraY(T23P) O-antigen blot is significantly less intense than the wildtype or vector. It would be nice to see images of the three full gels/blots in the supplementary material.

RESPONSE: We have included a new supplementary figure panel (Figure S6B) containing the Western blots for the three biological replicates probing for O-antigen. In each blot, a modest decrease in O-antigen levels is detected in the cells producing MraY(T23P). Densitometry measurements may not be ideal as the reviewer points out, but there are not any other options for quantification in this case. The quantification shows there is a 30% decrease in O-antigen in *P. aeruginosa* cells expressing MraY(T23P) (Figure S6B, C). Such a reduction is also supported by our data showing the susceptibility of these cells to its self-produced R2 pyocin (Figure S6A), which requires a receptor that is only accessible when there is a reduction in O-antigen present on the cell surface. Thus, the combination of O-antigen quantification and the pyocin susceptibility results makes a strong case that O-antigen levels are reduced in cells producing MraY(T23P).

3. “Using course-grained MD simulations” on Line 291 should be changed to “coarse”.

RESPONSE: This has been changed. Thank you for catching this typo.

Reviewer #2 (Remarks to the Author):

The bacterial cell envelope is an essential structure that resists antibiotics, immune systems, and environmental hazards. Bacterial glycoconjugates are foundational for envelope assembly and include the peptidoglycan (PG) cell wall, LPS O-antigens, capsules, and other clinically important structures. The glycan moiety of nearly all bacterial glycoconjugates is assembled on a lipid carrier known as undecaprenyl phosphate (Und-P). Since most bacteria encode multiple Und-P-utilizing pathways, the pool of Und-P must be shared between pathways. But how? While the mechanisms that regulate Und-P distribution are not known, the present work by Marmont et al. indicates that one mechanism involves feedback inhibition. Briefly, the authors isolated suppressors encoding an altered MraY enzyme (T23P mutation) that restores growth to *P. aeruginosa* cells defective for PG biosynthesis. Subsequent work confirmed that MraY(T23P) restores growth to *P. aeruginosa*, as well as *E. coli*, cells defective for PG biosynthesis. A closer examination revealed that cells expressing MraY(T23P) accumulate PG intermediates (lipid II), indicating that MraY(T23P) is a hyperactive enzyme. Since MraY competes with other initiating glycosyltransferases for Und-P, the authors wondered if misregulated MraY activity would impede the production of other Und-P-dependent polymers. Indeed, O-antigen production was reduced in cells expressing mraY(T23P). This finding suggests that the kinetics of initiating glycosyltransferases like MraY (PG) or WbpL (O-antigen) play a major role in regulating Und-P distribution. Finally, structural studies indicated the extracytoplasmic side of MraY is the regulatory site for this enzyme and that lipid II binding likely induces conformational changes in MraY that affect its activity.

Overall, the manuscript is well-written, the experiments well-controlled, and the conclusions appropriate. The finding that O-antigen expression is reduced in cells expressing mraY(T23P) is especially intriguing. It would be interesting to know if hyperactive variants of WbpL reduce PG production in *P. aeruginosa* in a similar manner. More generally, the discovery that feedback inhibition alters competition for Und-P has important implications for understanding how cell envelope assembly is coordinated. I have only a few minor comments.

RESPONSE: We thank the reviewer for their enthusiasm for our work and suggestions for its improvement. We agree that it would be interesting to know if hyperactive variants of WbpL reduce PG synthesis via C55-P competition similar to the effect of MraY(T23P) on O-antigen synthesis. Using our MraY mutants as a guide, we performed sequence and structural

9alignments of WbpL and MraY and selected 6 candidate residues that might result in hyperactive WbpL enzymes when altered. Unfortunately, none of these mutations had any effect on PG synthesis that we could measure. WbpL and MraY are only 23% identical in sequence such that a candidate approach to making the desired hyperactive mutants in WbpL is apparently not feasible. We are currently developing a genetic screen to isolate hyperactive mutants of WbpL, but this screen and the subsequent biochemical analysis of WbpL activity will likely require extensive optimization. Such an investigation is therefore beyond the scope of the current manuscript.

Reversible reactions

Lines 432-437: Can the feedback mechanism help explain why initiating glycosyltransferases like MraY catalyze reversible reactions (e.g., PMID: 14324547, 3034883, 318640, 16237026)?

RESPONSE: The reviewer asks an interesting question. However, we do not have a new explanation for why initiating glycosyltransferases are reversible based on the feedback mechanism. The reversibility observed in the studies referenced all examine the enzyme in the context of isolated cellular membranes using substrates that are exogenously provided. However, in cells, the MraY reaction product lipid I is rapidly converted to lipid II by MurG resulting in low levels of lipid I in cells. Because of this, the forward reaction is expected to be thermodynamically favored because there is comparatively more MraY substrate (UDP-MurNAc-pep5, C55P) than product, driving the reaction in this direction. It is therefore unclear whether the reversibility of the MraY reaction or other similar transferase reactions is physiologically relevant.

On a related note, can the authors comment on whether they think their hyperactive variants of MraY are capable of catalyzing the reverse formation of lipid I?

RESPONSE: We have not tested the reverse reaction in our in vitro system, but do not see any reason why the hyperactive variants would affect the reversibility. As described above, we think the forward reaction of MraY is most relevant for PG synthesis in cells and therefore focused our efforts on that reaction.

Multicopy suppression

10In Figure 1, the authors show that *MraY*(T23P) suppresses the growth defect of cells lacking *PonB* and *LpoA*. Since *PonB* and *LpoA* are required for PG biosynthesis, it would be helpful to see the shape of these cells.

RESPONSE: We thank the reviewer for their suggestion. We have added a supplementary figure (new Figure S1) containing phase contrast micrographs of both wild-type *Pseudomonas aeruginosa* PAO1 and *P. aeruginosa* Δ *ponB* Δ *lpoA* expressing *MraY* or *MraY*(T23P). These cells were grown in the permissive condition (rich media, lysogeny broth) because the *P. aeruginosa* Δ *ponB* Δ *lpoA* strain is unable to grow in the restrictive condition (Vogel-Bonner minimal media, VBMM) unless expressing *MraY*(T23P). In these micrographs, wild-type *P. aeruginosa* expressing either *MraY*(WT) or *MraY*(T23P) looks comparable to cells expressing empty vector whereas in the *P. aeruginosa* Δ *ponB* Δ *lpoA* background, only expression of *MraY*(T23P) suppresses the observed cell morphology defects (blebbing and rounding of cells).

Figure 2 and S3

Is there a reason significance was not plotted in Figs. 2 and S3? It is a little confusing to understand significance as described in the legends of these figures.

RESPONSE: We thank the reviewer for noticing this omission. We have revised the figures to include the significance.

Reviewer #3 (Remarks to the Author):

The manuscript by Marmont and colleagues entitled “A feedback control mechanism governs the synthesis of lipid-linked precursors of the bacterial cell wall” provides new information on how bacterial cells balance the utilization of the precious resource C55P/PP. This lipid-based carrier molecule can be decorated with glycans for the subsequent use in vastly distinct components of the cell envelope. This includes periplasmic peptidoglycan, as well extracellularly polysaccharides in the form of O-antigen. Although there are likely to be multiple control points across various stages of these biochemical pathways, the authors have specifically interrogated when C55 is linked to a GlcNAc-MurNAc-pentapeptide (lipid II), the final precursors for peptidoglycan biosynthesis. The team of researchers capitalised on their previous findings on this pathway. In this study, they designed a series of assays to illustrate that that mutation of T23 in *MraY* leads to the protein being resistant to Lipid II accumulation, and the subsequent starvation of C55 in other cellular aspects, eg *Oag*. Using a combination of cryoEM and in silico analyses, the authors show that Lipid II binding in *MraY* occurs at the periplasmic face of the IM,

11and that the T23P mutation is likely to obliterate this binding capacity. Since the structural analyses indicate that binding of flipped (periplasmic facing) Lipid II can lead to allosteric modification in the protein's active site, the T23P mutant retains activity, despite Lipid II hyper-accumulation.

Comments:

The phenotypic analyses are somewhat indirect. Additional whole cell assays to back up the findings in Ec and Pa are desired. Direct co-purification of Lipid II-WT *MraY* vs Lipid II-*MraY*T23P would provide stronger evidence.

RESPONSE: Evidence that *MraY* directly binds Lipid II has already been reported

(Oluwole et al 2022). We have performed a similar analysis for both *MraY*(WT) and *MraY*(T23P) and have also observed Lipid II in the purified preparations. Lipid II co-purified with both *MraY* proteins, but the amount was variable from preparation to preparation. We therefore did not include these results. The structural analysis of both *MraY*(WT) and *MraY*(T23P) show density at the dimer interface consistent with Lipid II. Also, the MD analysis suggests that both proteins also can bind Lipid II. Together with the prior study showing Lipid II binding, we think the data strongly support Lipid II occupying the cavity in the dimer interface in WT and activated forms of the protein. We therefore favor a model in which Lipid II associates with both *MraY*(WT) and *MraY*(T23P) but that the activated variant has activity that is insensitive to this binding. We have made this point more explicitly in the revised Discussion section.

Alternative means of illustrating the significance of the *MraY* periplasmic lipid II binding requirements, other than mutation of *ponA/B*/*lpoA/B* are desired. For example, specific conditions (eg antimicrobials) that alter PG vs Oag demands will place a greater emphasis on the biological dynamics of this feedback loop.

RESPONSE: Unfortunately, there are not many straightforward ways to alter the demands for C55P by the PG or O-Ag pathways in cells, especially gram-negative bacteria that exclude many antibiotics. We think the mutant strains used address the issue at hand better than antibiotics because they are more specific and do not require assumptions about cell penetration like antibiotics do. Nevertheless, we were able to add two additional experiments that further support the idea that mutant phenotypes are a result of altered demand on the C55P pool.

In the first experiment, the initiator of O-Ag synthesis, WbpL, was overproduced in the *P. aeruginosa* Δ ponB Δ lpoA strain expressing *mraY(T23P)* from the chromosome (the original isolated suppressor strain). This overproduction reduced the suppressing activity of the *mraY(T23P)* mutation for growth on VBMM medium, suggesting that an increase in WbpL activity reroutes C55P towards the production of O-antigen, thereby reducing the amount of available C55P for lipid II synthesis. This result supports the idea that cells making *MraY(T23P)* shift more C55P towards lipid II and that this effect can be countered by enhancing O-Ag synthesis. This data is now included as new Figure S6D.

The second experiment investigated sensitivity to the beta-lactam carbenicillin. Beta-lactams target the penicillin-binding proteins (PBPs) and block the crosslinking of nascent PG glycans into the mature matrix. In prior work (Cho et al. 2014), we showed that these uncrosslinked glycans are rapidly degraded following beta-lactam treatment, resulting in a futile cycle of PG synthesis and degradation that contributes to beta-lactam toxicity. Under these conditions, there is a high demand for lipid II that may be difficult to meet due to the *MraY* feedback loop. Accordingly, cells producing *MraY(T23P)* grow better in the presence of carbenicillin relative to those producing *MraY(WT)*, further supporting a role for *MraY* regulation in controlling lipid II levels. This data is now included as new Figure S4.

A competitive MD analysis of its cytoplasmic enzymatic role, and Lipid II periplasmic sequestering may shed light on the proposed direct inhibition model.

RESPONSE: In our simulations, we see enhanced dynamics of the C-terminal end of TM9 when lipid II is bound relative to the unbound state. This end of the helix is on the cytoplasmic side of the enzyme such that the enhanced dynamics here provide a potential means by which lipid II binding alters enzymatic activity in the nearby active site. The additional analysis suggested by the reviewer would be interesting to try. However, on the nanosecond timescale at which our MD analysis is performed, we are unlikely to see changes that would be any more informative of the mechanism than what we have observed already.

Line 425-437 are weak, thereby impacting the relevance of Line 87-97. Expansion of knowledge

by including bioinformatics on more species/across biology is appropriate if the authors want to retain the broader significant claims in the introduction and discussion.

RESPONSE: We respectfully disagree with the reviewer on this comment. Lines 87-97 describe the general problem of controlling C55P usage among competing pathways, which is required background information relevant to our work with MraY. We are not making any claims in this section. Regarding lines 425-437 in the Discussion section, we think the section appropriately puts our work in a broader perspective by suggesting that other similar enzymes may share this regulatory mechanism. In our opinion, the purpose of a Discussion section is to present readers with plausible extensions of current results to inspire future studies. We think our suggestion that other similar enzymes may share a regulatory mechanism with MraY is an entirely reasonable one such that we would be remiss not to mention it.

If there were a bioinformatic analysis that would clearly indicate whether or not a particular enzyme shared the feedback mechanism with MraY, we would be excited to try it. However, we do not currently have a sequence signature or set of residues that would be predictive nor do we think one is likely given the diversity of lipid-linked products formed by the different glycan biogenesis pathways that would serve as feedback inhibitors.

Line 188: Normalization by optical density is not adequate as the sole means to define inputs, as these values can be influenced by many factors which would skew the data (eg cell size/morphology, capsule production). Please combine with eg CFU, microscopy, protein quant.

RESPONSE: The reviewer is technically correct. Optical density can be influenced by factors like cell shape etc. However, in our extensive experience making cell extracts from cells derived from the same parent strain background, the differences are minor. We often make extracts from normally dividing cells in parallel with filamentous (non-dividing) cells based on OD₆₀₀ and find that, despite the great differences in cell shape, the protein concentrations of the resulting extracts are nearly identical. In our case, the cells collected for analysis have been imaged and have largely similar morphologies (see new Figure S1). Also, the colonies of the various strains used do not show any major differences in mucoidy indicative of changes in capsule production. It is therefore highly unlikely that the major changes we observe in lipid II levels (2-4x increase) are due the slight variances in optical density between cultures. We are thus confident in the lipid II quantification as presented.

Line 358: It levels off in the WT, that is a measure of activity and then loss of activity, hence the inability to inhibit WT MraY activity needs to be addressed.

RESPONSE: The reviewer is correct that there is a small window of activity for MraY(WT) before it levels off. However, this level is very low such that showing loss of this activity with added lipid II would only be a modest benefit. Furthermore, even if we saw loss of activity, it would be difficult to distinguish between feedback inhibition and binding competition with C55P substrate. The mechanism that we propose requires proper orientation of the lipid II in the membrane, which we cannot control in an in vitro system. There are therefore limitations to what can be tested directly with biochemistry, not to mention the high degree of difficulty of working with a polytopic membrane protein and a potential inhibitor that is very hydrophobic and difficult to obtain/purify. If we were working with a soluble enzyme with a soluble inhibitor, we would agree with the reviewer that this experiment would be an important one to try. But, given the argument above and the totality of the data presented, we hope the reviewer agrees that we've made the strongest possible case we can at this point to support our model even if some of the evidence is necessarily indirect given the role of membrane orientation in the regulatory system.

Line 381- 406: These sections appear more appropriate for the results, not discussion.

RESPONSE: The passage referred to by the reviewer indeed describes results. However, the text is needed to summarize results to provide context for the discussion points that follow.

Figure 3: Please discuss the role of K166, as it is not directly involved in binding, could it be Lipid II recruitment?

RESPONSE: The reviewer raises an interesting question. Although it is not involved in long lasting interactions (which we have defined as >60% of the simulation time), K166 is peripherally involved in the interaction with lipid II (see Figure S9). Given the flexibility of the lipid II molecule (Figure S10), K166 may play more of an auxiliary role in binding given the lipid II flexibility and the location and charge of this residue.

Are the hydrophobic residues lining the cavity highly conserved across *Pseudomonas* spp, or in

other genera?

RESPONSE: The specific hydrophobic residues lining the cavity aren't conserved, but the hydrophobic nature of the cavity is. In the revised manuscript, we have added a supplementary figure (new Figure S7) of the structures of Mray from *Escherichia coli*, *Aquifex aeolicus*, and *Enterocloster boltae*, along with the AlphaFold2 model of Mray from *Pseudomonas aeruginosa* that illustrate the conserved hydrophobic nature of the cavity.

Lines 90 & 91: Change “produce” to for example “carry”

RESPONSE: We have made the requested change.

Line 193: Why are Lipid II levels not higher in the E. coli mutant with MrayT23P overexpression? This section is very succinct, and details are missing.

RESPONSE: We thank the reviewer for being so thorough. In the analysis of the individual samples, the trend in each run was the same, with lipid II levels being least in the empty vector sample, intermediated with Mray(WT) production, and highest in cells making Mray(T23P). The quantities detected in this case were just a little more variable such that the increase upon Mray(T23P) production was not statistically significant. We have added this information to the figure legend.

Line 228: “Strikingly” is an overstatement.

RESPONSE: We have revised the text.

Line 305: Only W217 is at 100%

RESPONSE: We have adjusted the text to reflect this.

Line 325: Reword “the first membrane step”

RESPONSE: We have reworded the text.

Figure 2: Embed stats in bar graphs.

RESPONSE: We have plotted the significance in the bar graphs in Figures 2 and S5.

Figure 4: Improve quality of the x- and y-labels.

RESPONSE: We have adjusted the font size and overall quality of the X- and Y-labels for clarity.

Decision Letter, first revision:

Message: Our ref: NMICROBIOL-23071944A

21st December 2023

Dear Dr. Bernhardt,

Thank you for your patience as we've prepared the guidelines for final submission of your Nature Microbiology manuscript, "A feedback control mechanism governs the synthesis of lipid-linked precursors of the bacterial cell wall" (NMICROBIOL-23071944A). Please carefully follow the step-by-step instructions provided in the attached file, and add a response in each row of the table to indicate the changes that you have made. Please also check and comment on any additional marked-up edits we have proposed within the text. Ensuring that each point is addressed will help to ensure that your revised manuscript can be swiftly handed over to our production team.

If you have not done so already, please alert us to any related manuscripts from your group that are under consideration or in press at other journals, or are being written up for submission to other journals (see: <https://www.nature.com/nature-research/editorial->

17policies/plagiarism#policy-on-duplicate-publication for details).

In recognition of the time and expertise our reviewers provide to Nature Microbiology's editorial process, we would like to formally acknowledge their contribution to the external peer review of your manuscript entitled "A feedback control mechanism governs the synthesis of lipid-linked precursors of the bacterial cell wall". For those reviewers who give their assent, we will be publishing their names alongside the published article.

Nature Microbiology offers a Transparent Peer Review option for new original research manuscripts submitted after December 1st, 2019. As part of this initiative, we encourage our authors to support increased transparency into the peer review process by agreeing to have the reviewer comments, author rebuttal letters, and editorial decision letters published as a Supplementary item. When you submit your final files please clearly state in your cover letter whether or not you would like to participate in this initiative. Please note that failure to state your preference will result in delays in accepting your manuscript for publication.

Cover suggestions

COVER ARTWORK: We welcome submissions of artwork for consideration for our cover. For more information, please see our [a guide for cover artwork](https://www.nature.com/documents/Nature_covers_author_guide.pdf).

Nature Microbiology has now transitioned to a unified Rights Collection system which will allow our Author Services team to quickly and easily collect the rights and permissions required to publish your work. Approximately 10 days after your paper is formally accepted, you will receive an email in providing you with a link to complete the grant of rights. If your paper is eligible for Open Access, our Author Services team will also be in touch regarding any additional information that may be required to arrange payment for your article.

Please note that *Nature Microbiology* is a Transformative Journal (TJ). Authors may publish their research with us through the traditional subscription access route or make their paper immediately open access through payment of an article-processing charge (APC). Authors will not be required to make a final decision about access to their article until it has been accepted. [Find out more about Transformative Journals](https://www.springernature.com/gp/open-research/transformative-journals)

Authors may need to take specific actions to achieve [compliance with funder and institutional open access mandates](https://www.springernature.com/gp/open-research/funding/policy-compliance-faqs). If your research is supported by a funder that requires immediate open access (e.g. according to [18](https://www.springernature.com/gp/open-research/plan-s-Plan S principles) then you should select the gold OA route, and we will direct you to the compliant route where possible. For authors selecting the subscription publication route, the journal's standard licensing terms will need to be accepted, including self-archiving policies. Those licensing terms will supersede any other terms that the author or any third party may assert apply to any version of the manuscript.

For information regarding our different publishing models please see our Transformational Journals page. If you have any questions about costs, Open Access requirements, or our legal forms, please contact ASJournals@springernature.com.

Best regards,

Reviewer #1:

Remarks to the Author:

In this revised manuscript, Marmont et al. have addressed the minor points that I raised and I have no further concerns related to the publication of the study.

- Sean Workman, PhD

Reviewer #2:

Remarks to the Author:

The authors have satisfied my concerns.

Reviewer #3:

Remarks to the Author:

I congratulate the authors on an outstanding body of research. All my queries from the first round of review have been adequately addressed and I endorse publication of the manuscript.

Final Decision Letter:

Message: 11th January 2024

Dear Dr Bernhardt,

I am pleased to accept your Article "Synthesis of lipid-linked precursors of the bacterial cell wall is governed by a feedback control mechanism in *Pseudomonas aeruginosa*" for publication in Nature Microbiology. Thank you for having chosen to submit your work to us and many congratulations.

Please note that *Nature Microbiology* is a Transformative Journal (TJ). Authors may publish their research with us through the traditional subscription access route or make

20their paper immediately open access through payment of an article-processing charge (APC). Authors will not be required to make a final decision about access to their article until it has been accepted. [Find out more about Transformational Journals](https://www.springernature.com/gp/open-research/transformational-journals)

Authors may need to take specific actions to achieve [compliance with funder and institutional open access mandates](https://www.springernature.com/gp/open-research/funding/policy-compliance-faqs). If your research is supported by a funder that requires immediate open access (e.g. according to [Plan S principles](https://www.springernature.com/gp/open-research/plan-s-compliance)) then you should select the gold OA route, and we will direct you to the compliant route where possible. For authors selecting the subscription publication route, the journal's standard licensing terms will need to be accepted, including [self-archiving policies](https://www.nature.com/nature-portfolio/editorial-policies/self-archiving-and-license-to-publish). Those licensing terms will supersede any other terms that the author or any third party may assert apply to any version of the manuscript.

With kind regards,